# LSH mediates gene repression through macroH2A deposition

Kai Ni[1], Jianke Ren[1], Xiaoping Xu[1], Yafeng He[1], Richard Finney[2], Simon M. G. Braun[3], Nathaniel A. Hathaway[4], Gerald R. Crabtree ⬥ [3,5] & Kathrin Muegge ⬥ [1,6 ✉]

The human Immunodeficiency Centromeric Instability Facial Anomalies (ICF) 4 syndrome is a severe disease with increased mortality caused by mutation in the LSH gene. Although LSH belongs to a family of chromatin remodeling proteins, it remains unknown how LSH mediates its function on chromatin in vivo. Here, we use chemical-induced proximity to rapidly recruit LSH to an engineered locus and find that LSH specifically induces macroH2A1.2 and macroH2A2 deposition in an ATP-dependent manner. Tethering of LSH induces transcriptional repression and silencing is dependent on macroH2A deposition. Loss of LSH decreases macroH2A enrichment at repeat sequences and results in transcriptional reactivation. Likewise, reduction of macroH2A by siRNA interference mimicks transcriptional reactivation. ChIP-seq analysis confirmed that LSH is a major regulator of genome-wide macroH2A distribution. Tethering of ICF4 mutations fails to induce macroH2A deposition and ICF4 patient cells display reduced macroH2A deposition and transcriptional reactivation supporting a pathogenic role for altered marcoH2A deposition. We propose that LSH is a major chromatin modulator of the histone variant macroH2A and that its ability to insert marcoH2A into chromatin and transcriptionally silence is disturbed in the ICF4 syndrome.

[1] Mouse Cancer Genetics Program, National Cancer Institute, Frederick, MD 21702, USA. [2] CCR Collaborative Bioinformatics Resource, Center for Cancer Research, National Cancer Institute, Bethesda, MD 20892, USA. [3] Departments of Pathology and Developmental Biology, Stanford University School of Medicine, Stanford, CA 94305, USA. [4] Division of Chemical Biology and Medicinal Chemistry, Center for Integrative Chemical Biology and Drug Discovery, UNC Eshelman School of Pharmacy, Chapel Hill, NC 27599, USA. [5] Howard Hughes Medical Institute, Chevy Chase, MD 20815, USA. [6] Basic Science Program, Leidos Biomedical Research, Inc., Frederick National Laboratory for Cancer Research, Frederick, MD 21702, USA. ✉email: Kathrin.Muegge@nih.gov

Eukaryotic chromatin is built of arrays of nucleosomes that each consists of 147 bp DNA wrapped around an octamer of core histones[1–3]. The octamer is composed of a central tetramer of histones H3 and H4 flanked by two heterodimers of histones H2A and H2B. Chromatin dynamics is controlled by histone modifiers and chromatin remodeling complexes that alter chromatin structure and regulate nuclear processes such as transcription, replication, recombination, or DNA repair[4,5].

The canonical core histones can be replaced with histone variants, including macroH2A[6]. Apart from a linker and a macrodomain, macroH2A contains a histone domain that is ~65% identical to canonical H2A, but differs in the L1 loops which is important to stabilize DNA binding to the octamer[7]. Reconstitution of nucleosomes containing macroH2A renders them more resistant to exonuclease digestion and promotes interactions between nucleosomes[8]. This furthers chromatin condensation and contributes to the more compacted nature of heterochromatin[8,9]. MacroH2A is a family of variants, including macroH2A1.1 and macroH2A1.2 (isoforms transcribed from one gene) and macroH2A2, which are frequently associated with constitutive or facultative heterochromatin and transcriptional silencing[9–12]. However, macroH2A1 can also reside in euchromatin and positively influence transcription, a feature which is primarily attributed to the isoform macroH2A1.1[9,13,14]. Recently, it has been suggested that macroH2A may reinforce active or repressed expression states and buffer transcriptional noise[3,15]. Several factors have been identified that negatively regulate macroH2A incorporation in vivo, however, factors that control its deposition into chromatin remain unknown[9].

Patients with Immunodeficiency, Centromeric instability and Facial dysmorphism syndrome type 4 (ICF4) suffer from a plethora of symptoms including organ malformations, facial anomalies, mental retardation, delayed growth, genomic instability, and severe immunodeficiency which leads to a shortened lifespan[16,17]. The pathophysiology of ICF4 remains largely unknown, but the autosomal recessive disease is caused by a genetic mutation in *LSH/HELLS* (Lymphoid Specific Helicase, or HElicase Lymphoid Specific). *Lsh* knockout mice phenocopy many symptoms, and exhibit hematopoietic, neurologic, and germ cell deficiencies, reduced growth, and early lethality[18–23].

LSH belongs to the SNF2 family of ATP-dependent chromatin remodelers[1,24,25]. Members of the family are thought to translocate along DNA. This characteristic enables them to evict nucleosomes, slide nucleosomes along DNA or exchange histone variants. LSH can perform nucleosome sliding in vitro and its activity is enhanced by CDCA7, a protein that causes a related syndrome, known as ICF2, when it is mutated[26–28]. LSH recruitment results in reduced chromatin accessibility in vivo and LSH function is associated with transcriptional silencing[27–31]. LSH exerts its effects on repeat sequences embedded in heterochromatin and is associated with high DNA methylation levels[30,32].

While LSH function is associated with heterochromatin it remains unknown if and how it induces repressed chromatin states and whether this relates to its chromatin remodeling function. Here, we delineate the causal sequences of biochemical events induced by LSH using chemical induced proximity[33]. We define a role of LSH in macroH2A deposition and examine its potential role in ICF4 patients.

## Results

### LSH tethering induces transcriptional repression.
To define the primary molecular function of LSH, we applied a chemical-induced proximity (CIP) assay in which a fusion protein can be tethered to one genetically modified *Oct4* allele[34]. We utilized a previously generated murine embryonic stem (ES) cell line containing an array of DNA binding domains (12 × ZFHD1) upstream of the *Oct4* transcription start site (TSS) on one allele (Fig. 1a)[34,35]. Insertion of an in-frame eGFP gene into one *Oct4* allele serves as indicator of transcriptional regulation with GFP as transcriptional readout, whereby OCT4 can be only expressed by the unmodified wild-type allele. The genetically modified *Oct4* locus has been previously shown to undergo repression upon ES differentiation[34,35]. LSH was stably expressed as fusion protein with the rapamycin binding domain FRB-V5 tagged (LSH-FRB-V5). Furthermore, the DNA binding domain of ZFHD1 was stably expressed as a fusion protein with the rapamycin binding domain FKBP-HA tagged (ZnF-FKBP-HA). Addition of rapamycin to the cell culture elicits an interaction between the two fusion proteins and recruitment of the complex to the modified *Oct4* allele[34,35]. In contrast to direct fusion of a protein to the DNA binding domain, which produces a rigid conformation, the fast off rate of rapamycin allows for more freedom of activity and a normal mode of action, as has been shown for the BAF complex[36].

Expression of wild-type LSH and DNA-binding fusion proteins were monitored by western blot analysis (Fig. 1b) and immunofluorescence staining (Supplementary Fig. 1). Reporter-GFP expression was not affected in the absence of rapamycin as judged by flow cytometry analysis (Supplementary Fig. 2a, b). After rapamycin treatment wild-type LSH was tethered to the modified *Oct4* allele within 15–60 min as measured by chromatin immunoprecipitation (ChIP) followed by qPCR analysis (Fig. 1c). Upon rapamycin addition we found that wild-type LSH progressively decreased GFP protein expression (Fig. 1d, e) and mRNA expression within 1 day of culture (Supplementary Fig. 3b). Using ChIP analysis, we observed decreased RNA polymerase II binding at the tethered locus (Fig. 1f) suggesting that the decline of mRNA level was due to reduced transcription. Importantly, transcriptional repression upon rapamycin treatment was specific for the modified allele (Supplementary Fig. 3a), since the amount of endogenous *Oct4* mRNA and OCT4 protein derived from the unmodified wild-type allele was unscathed upon rapamycin treatment (Fig. 1e and Supplementary Fig. 3c). Also, the targeted ES cells retained good morphology (Supplementary Fig. 3d) and maintained expression of pluripotency markers genes *Nanog* and *Sox2* under rapamycin exposure (Supplementary Fig. 3e, f), as has been previously reported[34,36].

To determine whether chromatin remodeling is required for LSH molecular function, we analyzed a catalytically inactive version of LSH with a single amino acid substitution (K 237A) within its ATP binding site. This site is critical for ATP hydrolysis and chromatin remodeling activity of SNF2 like factors[27,37]. Expression of the ATP mutant was comparable to wild-type LSH (Fig. 1b and Supplementary Fig. 1) and addition of rapamycin elicited a similar time kinetic of tethering to the engineered *Oct4* locus (Fig. 1c). Remarkably, the ATP mutant protein was incapable to reduce GFP mRNA and protein expression or Pol II binding (Fig. 1d, f and Supplementary Fig. 3b), suggesting that the chromatin remodeling activity of LSH contributes significantly to its function in GFP repression. Our results demonstrate that LSH recruitment can induce transcriptional repression and that LSH function depends on ATP binding and/or hydrolysis.

### LSH recruitment induces macroH2A deposition.
To investigate the molecular mechanism of LSH mediated GFP repression, we determined chromatin changes induced by LSH tethering. The design of primer location allowed us to discern chromatin changes that were specific for the genetically modified allele and not the wild-type *Oct4* allele (Fig. 2a). We found that histone

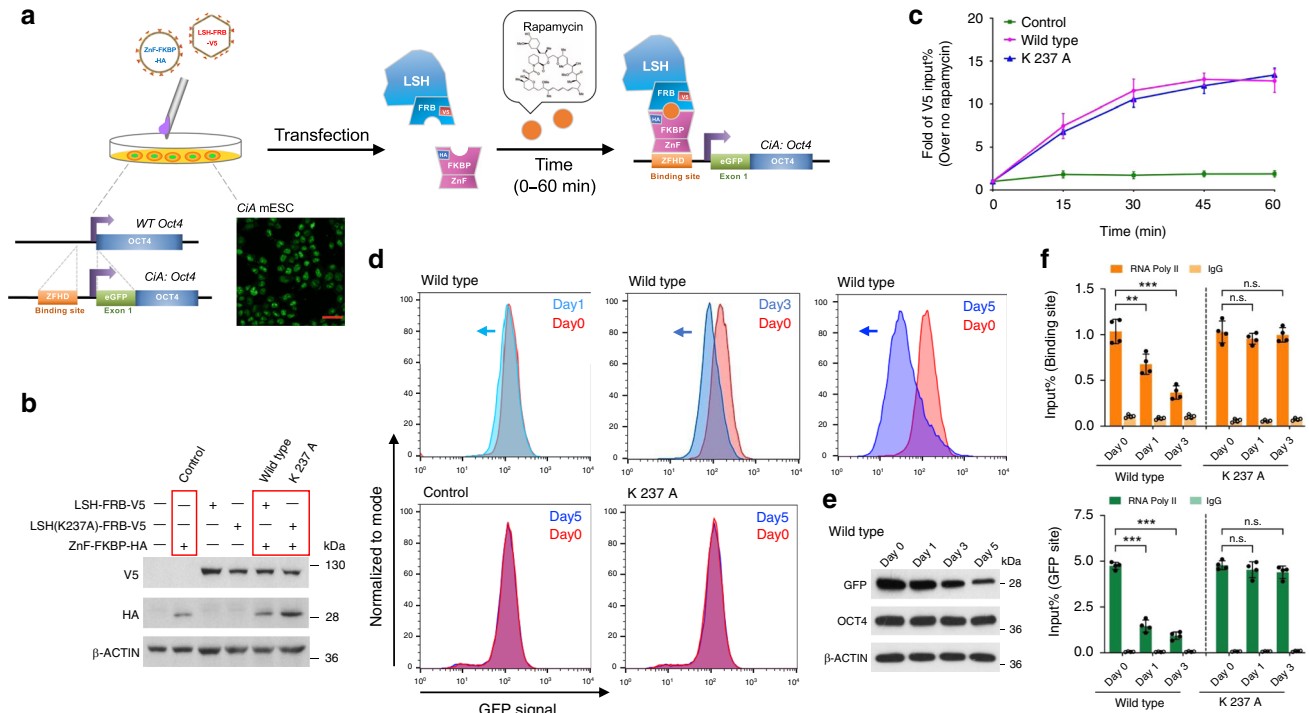

**Fig. 1 LSH tethering induces transcriptional repression. a** Schematic graph to illustrate recruitment of FRB-tagged LSH by rapamycin to the *CiA: Oct4* locus in the CIP system. Rapamycin dimerizes with FRB and FKBP. The *CiA: Oct4* mouse ES cells contain one modified *Oct4* allele harboring one array of DNA binding sites (12 × ZFHD1) in the promoter region upstream of an in-frame eGFP reporter inserted in exon 1 of the *Oct4* gene. Scale bar, 40 μm. **b** Western blot analysis for detection of LSH (wild type)-FRB-V5, LSH (K 237 A)-FRB-V5 (ATP mutant site), and ZnF-FKBP-HA fusion proteins in the engineered mouse ES cells using V5 and HA antibodies. Cells expressing both fusion proteins were used to establish the CIP system, and cells lacking LSH fusion protein served as controls. **c** ChIP-qPCR analysis using V5 antibody for detection of wild-type and ATP mutant LSH (K 237A) protein recruitment to the *Oct4* locus site within 60 min after rapamycin treatment in the CIP system ($n = 3$ independent experiments). Cells that lacked LSH fusion protein served as controls. Data are represented as mean ± SD. **d** Flow cytometry was used to measure reporter-GFP expression after recruitment of wild-type or ATP mutant LSH (K 237A)-FRB-V5 fusion protein at indicated time points in the CIP system. Cells that lacked LSH fusion protein served as controls. **e** Determination of GFP and OCT4 protein level by WB after rapamycin treatment at indicated time points in the wild-type LSH CIP system. **f** ChIP-qPCR analysis to determine Pol II enrichment at the ZFHD binding site and GFP reporter site at indicated time points in the wild-type and ATP mutant LSH (K 237A) CIP system after rapamycin treatment ($n = 4$ independent experiments). ChIPs using IgG served as the controls. Data are represented as mean ± SD. Significance assessed using one-way ANOVA with Tukey's multiple comparison test (**adjusted $p = 0.0022$, ***adjusted $p < 0.0001$, n.s. means not significant). Source data are provided as a Source Data file.

modifications closely associated with transcription were altered in dependence of LSH function. The 'active' chromatin markers H3K4me3 and H3K27ac were gradually reduced after rapamycin addition within 1–3 days after rapamycin addition (Fig. 2b, c). Similarly, H3K4me2 and H3K14ac decreased after rapamycin treatment (Supplementary Fig. 4b, c). As expected, only the wild-type LSH but not the ATP mutant LSH (K 237A) form was capable to evoke these changes indicating a close association of histone modifications with transcriptional changes. In search for the emergence of repressive marks, we first examined H3K27me3 and H3K9me3 which are known to form large repressive domains at the TSS of *Oct4* and other silenced loci in terminally differentiated cells[36,38]. However, neither histone modifications appeared at the locus after LSH recruitment, although these marks were readily detected at the MyoD promoter, serving as a positive control (Fig. 2d and Supplementary Fig. 4d). In addition, CpG methylation, a mark appearing upon long-term repression[39] was not detected at the locus (Fig. 2e), which was consistent with the previous observations that H3K9 methylation precedes DNA methylation at the *Oct4* locus, and that it occurs only in embryonic stem cells upon differentiation[29,39].

To investigate the role of linker histones and histone variants, we first examined the presence of H1, since H1 family members are known to play a key role during hormonal gene repression[40].

However, we detected very little H1.2 enrichment at the locus (Fig. 2f) despite successful detection at satellite sequences used as a positive control as previously reported[41,42]. Nonetheless, the profile of H1.2 distribution was unchanged after rapamycin addition (Fig. 2f). In contrast, the enrichment profiles of macroH2A underwent substantial changes upon LSH recruitment. MacroH2A is a histone variant which renders nucleosomes more stable and less accessible to exonuclease digestion, and which is closely linked to gene repression, heterochromatin, and chromatin compaction[7,9,43,44]. Significant increases of macroH2A1 as well as macroH2A2 were observed within 1 day after rapamycin addition (Fig. 2g, h). Since we found that ES cells expressed predominantly macroH2A1.2 (Supplementary Fig. 4a), consistent with previous reports[13,45–47], it suggested that LSH was capable to mediate the deposition of macroH2A1.2 and macroH2A2. Deposition of H2A was slightly reduced, suggesting an exchange between the canonical histone and its variant upon LSH recruitment (Fig. 2i). To test for histone variant selectivity, we investigated H2A.Z enrichment since it has been associated with transcriptional changes[48]. However, we did not find any significant changes in H2A.Z occupancy (Supplementary Fig. 4e). As a further control, we examined the presence of H3 and found no change in its distribution profile upon rapamycin treatment (Supplementary Fig. 4f). Importantly, we confirmed that changes

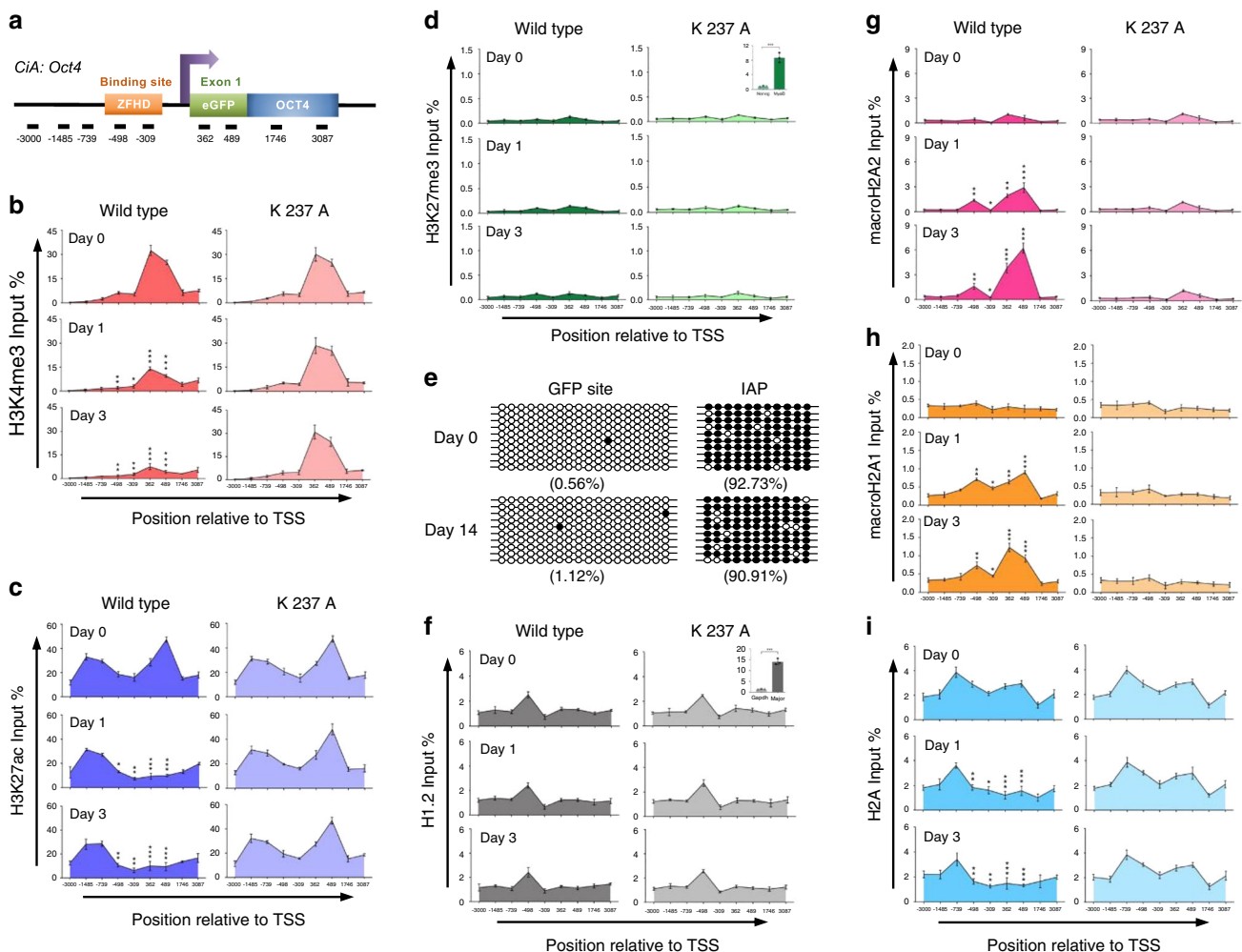

**Fig. 2 LSH recruitment induces macroH2A deposition at the tethered locus. a** Schematic representation of primer location at the *CiA: Oct4* allele; allele-specific primers at ZFHD binding (−498, −309) and eGFP sites (362, 489), and common primers of the *Oct4* gene (−3000, −1485, −739, 1746, and 3087). **b–d** ChIP-qPCR analysis to assess the dynamic changes of histone modifications H3K4me3 (**b**), H3K27Ac (**c**), and H3K27me3 (**d**) at the *CiA: Oct4* locus in the wild-type and ATP mutant LSH (K 237A) CIP systems after 0, 1, and 3 days of rapamycin treatment. *Nanog* and *MyoD* genes were used for H3K27me3 ChIP-qPCR results as negative and positive controls, respectively. *adjusted $p < 0.05$, **adjusted $p < 0.01$, and ***adjusted $p < 0.001$. **e** Bisulfite sequencing analysis shows no changes of CG methylation level at eGFP site following LSH recruitment by rapamycin for 14 days. IAP repeat sequence served as positive control for bisulfite sequencing analysis. The CpG methylation profiles are shown with black (methylated) and white (unmethylated) circles. Each sample included 10 sequenced clones and is represented as percentage of methylated CpGs. **f–i** ChIP-qPCR analysis to assess the dynamic changes of histones and variants H1.2 (**f**), macroH2A2 (**g**), macroH2A1 (**h**), and H2A (**i**) at the *CiA:Oct4* locus in the wild-type and ATP mutant LSH (K 237A) CIP systems after 0, 1, and 3 days of rapamycin treatment. *Gapdh* and *Major* satellite genes were used as negative and positive controls for H1.2 ChIP-qPCR result. *adjusted $p < 0.05$, **adjusted $p < 0.01$ and ***adjusted $p < 0.001$. One-way ANOVA with Tukey's multiple comparison test (**b–d**, **f–i**). **b–d**, **f–i** representative of three independent experiments. Data are represented as mean ± SD. Source data are provided as a Source Data file.

in macroH2A enrichment upon LSH tethering depended on the presence of the catalytic ATP binding site, since the ATP mutant LSH (K 237A) was incapable of inducing macroH2A deposition or altering H2A occupancy (Fig. 2g–i). This implies that ATP function is required for LSH-mediated macroH2A1.2 and macroH2A2 accumulation.

**LSH depletion reduces macroH2A level at repeat sequences**. To explore whether LSH affects macroH2A deposition under physiologic conditions, we examined macroH2A occupancy at genomic sites of known enrichment, including genomic sequences of constitutive heterochromatin and ribosomal DNA (rDNA). We used *Lsh*−/− and *Lsh*+/+ MEFs (Fig. 3a)[12,49], since they expressed higher levels of macroH2A1.2 compared to ES cells and since differentiated cells are known to express more macroH2A2

compared to ES cells (Supplementary Fig. 4a)[13,45–47]. Using ChIP followed by qPCR analysis, we found less macroH2A1 and macroH2A2 incorporation in LSH deficient cells at several repeat sequences including LINE1, IAP, and satellite sequences, whereas H2A occupancy was slightly increased (Fig. 3b). Likewise, the rDNA locus displayed high enrichment in LSH proficient cells, but reduced level of macroH2A1 and macroH2A2 in LSH-deficient MEFs (Fig. 3c, d). Moreover, we found LSH enrichment at the same regions of high macroH2A occupancy in *Lsh* wild-type MEFs indicating that LSH and macroH2A can localize to the same genomic regions (Fig. 3b, d). To test whether LSH modulates macroH2A deposition in other cell types, we depleted LSH by sh*LSH* RNA interference in human U2OS cells (Supplementary Fig. 5a), which express predominantly macroH2A1.2 (Supplementary Fig. 4a)[13,45–47]. We observed a similar reduction

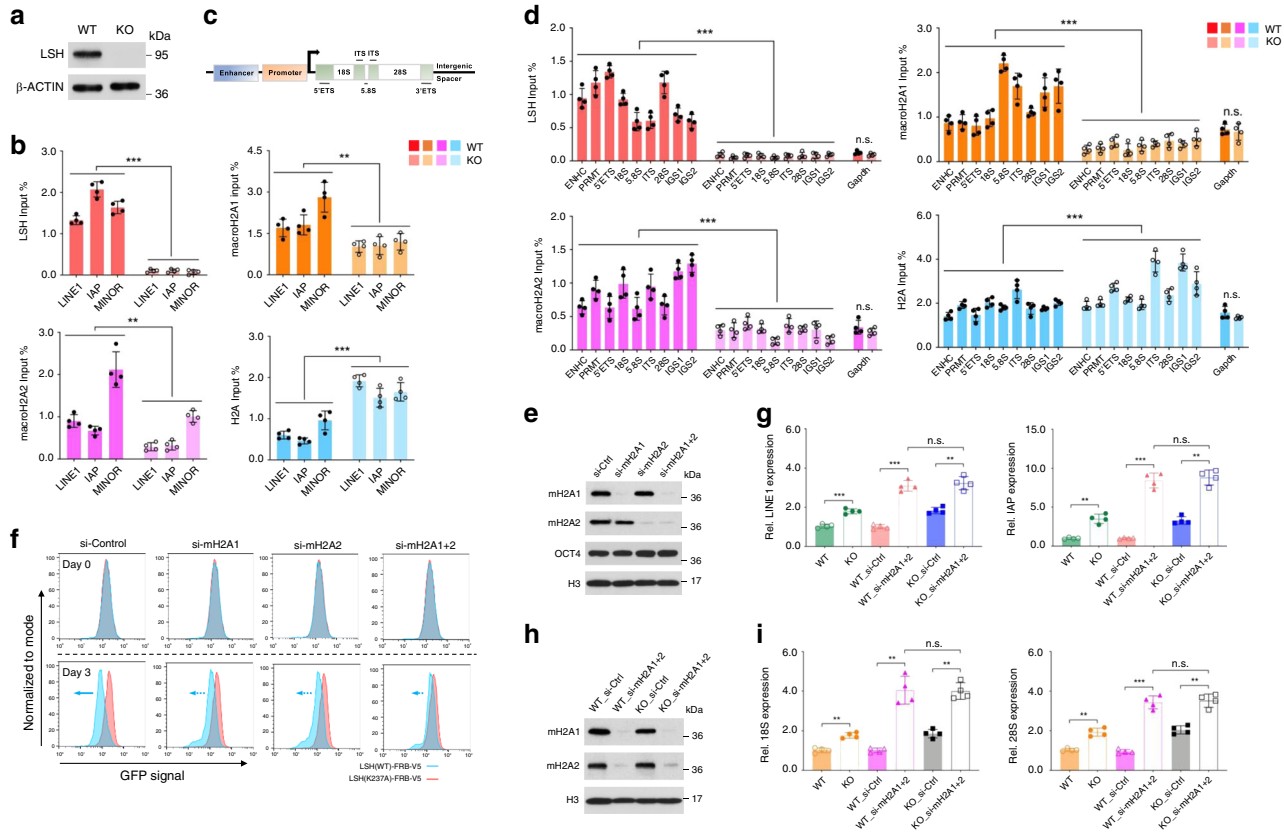

**Fig. 3 LSH induced transcriptional repression is in part mediated by macroH2A. a** Western blot analysis for detection of LSH protein in *Lsh* WT and KO MEFs. **b** ChIP-qPCR analysis for detection of LSH (***$p < 0.0001$), macroH2A1 (**$p = 0.0002$), macroH2A2 (**$p = 0.0002$), and H2A (***$p < 0.0001$) enrichment at repeat sequences, including LINE1, IAP, and MINOR satellites, in KO MEFs compared to WT cells. **c, d** Schematic representation of mouse rDNA repeats (**c**). ENHC enhancer, PRMT promoter, ETS external transcribed spacer, ITS internal transcribed spacer, IGS intergenic spacer. ChIP-qPCR analysis for detection of LSH, macroH2A1, macroH2A2 and H2A enrichment at rDNA sequences in KO MEFs compared to WT MEFs (**d**). ***$p < 0.0001$, n.s. means not significant. **e** Western blot analysis to detect indicated proteins after single or combined depletion of macroH2A1 and macroH2A2 using siRNA in the wild-type LSH CIP system. Depletion of macroH2A histones did not affect *Oct4* gene expression. **f** Flow cytometry to assess reporter-GFP expression in the wild-type LSH (blue) CIP system treated with control siRNA (si-Ctrl), macroH2A1 siRNA (si-mH2A1), macroH2A2 siRNA (si-mH2A2), or two combined siRNA (si-mH2A1 + 2) after 0 and 3 days of rapamycin addition. ATP mutant LSH (K 237A, red) CIP system was used as a negative control. **g–i** RT-qPCR analysis in *Lsh* WT and KO MEFs, or WT and KO MEFs treated with control siRNA (si-Ctrl) or two combined macroH2A siRNA (si-mH2A1 + 2) to compare the effects of LSH and macroH2A depletion on repeats (LINE1 and IAP, **g**) and rDNA (18S and 28S, **i**) transcription levels. Depletion of macroH2A proteins was confirmed by western blot analysis (**h**). **adjusted $p < 0.01$ and ***adjusted $p < 0.001$, n.s. means not significant. Data are represented as mean ± SD. Paired two-tailed Student's *t* test (**b, d**); one-way ANOVA with Tukey's multiple comparison test (**g, i**). **b, d, g, i** representative of four independent experiments. Source data are provided as a Source Data file.

of macroH2A1.2 and macroH2A2 at diverse repeat sequences, including satellite sequences and LINE1 elements (Supplementary Fig. 5b) and at ribosomal genes (Supplementary Fig. 5c, d). In addition, we detected LSH occupancy in U2OS cells at the same regions where macroH2A was enriched indicating a certain degree of co-localization (Supplementary Fig. 5b, d). Our results suggest that loss of LSH decreases macroH2A enrichment at several loci including ribosomal genes and some repeat sequences.

**Transcriptional repression is in part mediated by macroH2A.** To investigate whether LSH mediated transcriptional silencing depends on macroH2A1.2 or macroH2A2 deposition, we depleted macroH2A in the genetically modified ES cells by siRNA interference (Fig. 3e). After single depletion of either macroH2A1 or macroH2A2, we observed only partial GFP repression upon rapamycin treatment (Fig. 3f). Notably, combined depletion of macroH2A1 and macroH2A2 was most effective to abrogate rapamycin induced GFP repression (Fig. 3f). This suggests that LSH-mediated transcriptional repression required macroH2A and that both histone variants contributed to silencing.

LSH is critical to repress several repeat sequences embedded in heterochromatin[30,32,50]. Therefore, we analyzed repeat sequences that exhibited a decrease in macroH2A enrichment after LSH depletion (Fig. 3b and Supplementary Fig. 5b) and found that LINE1 or IAP elements were transcriptionally de-repressed in the absence of LSH in MEFs, and SAT2 and LINE1 in U2OS cells (Fig. 3g and Supplementary Fig. 5e). Likewise, depletion of macroH2A mimicked the LSH-deficient phenotype and led to an increase of LINE1 and IAP transcripts in MEFs (Fig. 3g, h), and SAT2 and LINE1 in U2OS cells (Supplementary Fig. 5e, f). In addition, 18S and 28S rDNA transcription, which is known to be suppressed by macroH2A[49], was enhanced in LSH or in macroH2A depleted MEFs and U2OS cells (Fig. 3i and Supplementary Fig. 5g). To understand whether LSH induced transcriptional repression is primarily mediated by macroH2A, we combined macroH2A depletion with LSH depletion (Fig. 3h and Supplementary Fig. 5f). We found no further additive effect in double depleted cells compared to macroH2A deficient cells at either of these repeat sequences and rDNA (Fig. 3g, i and Supplementary Fig. 5e, g). Our findings indicate that macroH2A

knockdown phenocopied LSH depletion and support a model in which LSH mediates repression via macroH2A deposition.

Since transcriptional pruning[12] is a mechanism that reduces maroH2A at highly transcribed genes, we wanted to examine whether changes of macroH2A in LSH deficient cells depend on changes in transcriptional activity. We selected several genes that are known to be silent in LSH proficient as well as LSH-deficient MEFs (Supplementary Fig. 6a)[30]. Also, it had been reported that these genes exhibit high macroH2A enrichment at their gene body in wild-type cells[12]. Thus, we tested whether macroH2A would be reduced at those sites in the absence of LSH using ChIP followed by qPCR analysis. All selected sites displayed significant decreases in macroH2A1 and macroH2A2 in LSH deficient cells compared to wild-type controls (Supplementary Fig. 6h) despite their silent state with or without LSH[30]. This suggested that reduction of macroH2A occupancy is insufficient to activate these genes and that multiple silencing mechanisms may be involved. Our observations are consistent with previous studies that report that heterochromatic genes remained silent upon macroH2A reduction and which proposed a hypothesis of redundant layers of silencing mechanism for macroH2A[9,51–53]. Importantly, this data also demonstrated that a decrease of macroH2A deposition in LSH deficient cells did not necessarily depend on transcriptional changes and that at those genomic location the presence or absence of LSH was likely the determining factor of macroH2A occupancy.

**Widespread changes in macroH2A deposition upon LSH depletion.** To determine the extent of macroH2A deposition in dependence of LSH, we first assessed the proportion of macroH2A associated with chromatin in LSH deficient cells. A previous study had reported that a decrease of H2A.Z chromatin association can serve as sign of reduced chromatin incorporation in cells deficient of the H2A.Z exchange factor EP400[54]. While total macroH2A protein levels were unaltered as judged by western blot analysis, macroH2A1 and macroH2A2 associated with chromatin were reduced by ~40% in LSH deficient cells compared to control cells (Fig. 4a and Supplementary Fig. 7a). This suggested a broad effect of LSH deletion on incorporation of macroH2A into chromatin.

Chromatin configuration and chromatin compaction may impact nuclear size and morphology[55]. We and others have noted that depletion of LSH can increase the nuclear size[56]. Using

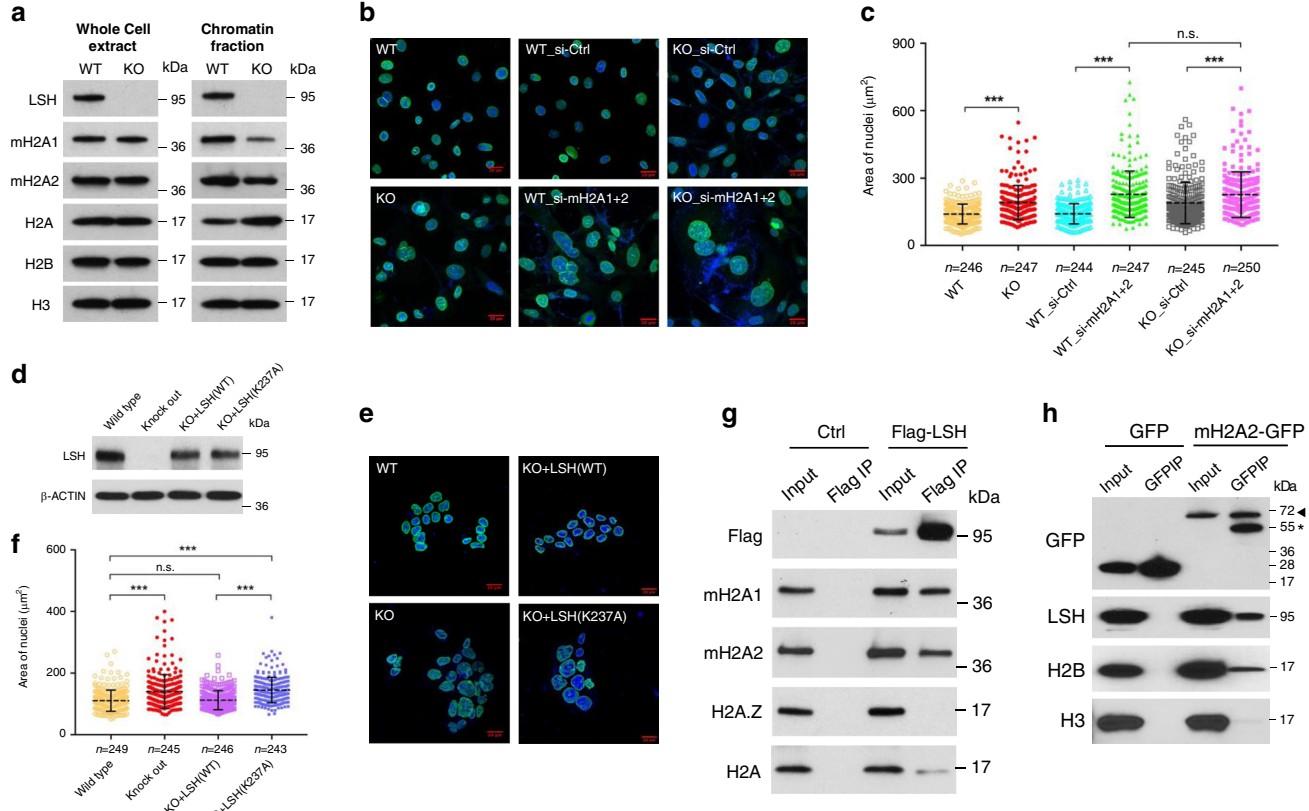

**Fig. 4 LSH depletion leads to decreased chromatin association of macroH2A and to an increase of nuclear size. a** Western blot analysis for detection of indicated proteins in whole-cell extract or in the chromatin fraction of *Lsh* WT and KO MEFs. **b, c** Microscopic images to visualize the nuclear size of *Lsh* WT and KO MEFs, or WT and KO MEFs treated with control siRNA (si-Ctrl) or two combined macroH2A siRNA (si-mH2A1 + 2) by staining with Lamin B1 and DAPI (**b**). Scale bar, 20 μm. Quantification of nuclear area of samples as shown in **c**. ***adjusted *p* < 0.0001, n.s. means not significant. **d** Western blot analysis to detect restoration of LSH protein level in *Lsh*−/− (KO) murine ES cells after transfection with wild-type LSH (WT) or ATP mutant LSH (K 237A) vector compared to *Lsh* WT and KO mouse ES cells. **e, f** Microscopic images to visualize the nuclear size in *Lsh* WT and KO mouse ES cells, and KO ES cells restored with wild-type LSH (WT) or ATP mutant LSH (K 237A) by staining with Lamin B1 and DAPI (**e**). Scale bar, 20 μm. Quantification of nuclear area of samples is shown in **f**. ***adjusted *p* < 0.0001, n.s. means not significant. **g** Flag-IP of indicated proteins confirmed by western blot analysis in U2OS cells with (Flag-LSH) or without (Ctrl) stable expression of Flag-LSH fusion protein. **h** GFP-IP of indicated proteins confirmed by western blot analysis in U2OS cells with a stably integrated vector for GFP or macroH2A2-GFP expression. An arrowhead indicates macroH2A2-GFP protein and an asterisk indicates a nonspecific cross-reaction band. Data are represented as dot plots with mean ± SD (**c, f**). One-way ANOVA with Tukey's multiple comparison test (**c, f**). Source data are provided as a Source Data file.

lamin B1 immunostaining and DAPI, we observed that the nuclear area was significantly enlarged in LSH depleted MEFs (Fig. 4b, c) or LSH deficient U2OS cells (Supplementary Fig. 7b, c). Furthermore, depletion of macroH2A was similarly leading to cells with a greater nuclear area (Fig. 4b, c and Supplementary Fig. 7b, c), as was previously reported[57]. To understand whether LSH induced an increase in the nuclear size through macroH2A, we combined macroH2A depletion with LSH deficiency in MEFs and U2OS cells. We found no further additive effect of LSH depletion in macroH2A deficient cells consistent with the notion that LSH mediated its effect on the nuclear size via macroH2A (Fig. 4b, c and Supplementary Fig. 7b, c). A similar increase in nuclear size was observed in $Lsh-/-$ (KO) ES cells compared to wild-type ES cells (Fig. 4d–f). To determine whether the effect of LSH was dependent on ATP function, we reconstituted $Lsh$ KO ES cells with either wild-type LSH or ATP mutant LSH (K 237A) expression vector (Fig. 4d). Only wild type but not the ATP mutant form of LSH was capable to restore the nuclear size implying ATP hydrolysis in the LSH dependent nuclear organization (Fig. 4e, f). The depletion of macroH2A in ES cells phenocopied LSH deficiency, and LSH depletion did not show an additive effect to macroH2A depletion supporting further the notion that the effect of LSH on nuclear structure is mediated by macroH2A deposition (Supplementary Fig. 7d–f).

To gain a better understanding, whether LSH is directly or rather indirectly responsible for macroH2A deposition, we tested for biochemical interactions between LSH and macroH2A using co-immunoprecipitation (CO-IP). Flagged-tagged LSH readily pulled down macroH2A1 as well as macroH2A2, but less so canonical H2A (Fig. 4g). Furthermore, the reverse approach was successful and GFP-tagged macroH2A2 could pull down LSH and H2B (Fig. 4h). Importantly, LSH did not coimmunoprecipitate the histone variant H2A.Z, indicating some degree of selectivity for LSH interactions (Fig. 4g). This result was also in agreement with the finding that LSH could not alter H2A.Z occupancy at the $Oct4$ locus upon rapamycin treatment (Supplementary Fig. 4e). These observations are consistent with a model in which LSH acts selectively and maybe directly involved in a process that promotes macroH2A deposition.

To assess genome wide distribution of macroH2A enrichment in dependence of LSH, we conducted ChIP-seq analysis in $Lsh-/-$ and $Lsh+/+$ MEFs that express predominantly macroH2A1.2 and macroH2A2 (Supplementary Fig. 4a). The UCSC genome browser view displayed areas of reduced macroH2A1 (three biologic replicas from three independently derived MEF cell lines for each genotype) and macroH2A2 accumulation (also three replicas) indicating widespread changes in the absence of LSH at some genomic locations (Fig. 5a). In contrast, ChIP-seq analysis of H2B was unaltered in the absence of LSH indicating an effect specific for macroH2A (Fig. 5a). In addition, we found that half a dozen of genes, for which we had earlier found differences of macroH2A enrichment by qPCR analysis corresponded well with the macroH2A-ChIP-seq data (Supplementary Fig. 6b–h). Since the distribution of macroH2A1 and macroH2A2 nucleosomes display a high degree of similarity[12,53] and since we found a very strong Pearson correlation (based on 5 Kb tiles across the genome) comparing macroH2A1 to macroH2A2 patterns in wild-type cells (Supplementary Fig. 8a), we assessed LSH mediated changes collectively as macroH2A. About 19.7% of the genome (divided in fixed 5 Kb tiles) displayed significantly altered macroH2A enrichment and 98.8% of these changes exhibited reduced macroH2A in LSH deficient cells (Fig. 5b). One reason for the large apparently unaffected compartment, may lie in macroH2A occupancy itself which is lower in the unaffected fraction of the genome compared to the affected (Supplementary Fig. 8b). To understand whether LSH

occupancy may determine the locations of change and whether LSH and macroH2A co-localize, we examined previously published LSH-ChIP-seq data[58]. Validation of the data set at sites with high and low LSH occupancy showed good correspondence between LSH-ChIP-seq data and LSH-ChIPs followed by qPCR analysis (Fig. 5c, d). Moreover, the vetted sites displayed a fair degree of concordance between LSH and macroH2A enrichment (Fig. 5c, d). Furthermore, the compartment of the genome which did not show significant changes of macroH2A enrichment (Fig. 5b), displayed significantly lower LSH enrichment compared to the genomic compartment that was affected (Supplementary Fig. 8c) suggesting that LSH occupancy was associated with changes of macroH2A occupancy. In addition, the genome wide Pearson correlation of macroH2A enrichment and LSH occupancy (based on 5Kb tiles) was $R = 0.419$ (untransformed data) and $R = 0.680$ ($log_2$ transformed data) indicating moderate to strong association between LSH occupancy and macroH2A deposition in the genome (Fig. 5e). As discussed below, a transcriptional pruning mechanism[12] may reduce the correlation strength between LSH and macroH2A at gene rich regions (Supplementary Fig. 8e). Nevertheless, these findings are consistent with a model that suggest a direct role for LSH in macroH2A deposition, but it does not exclude that other factors contribute as well to the distribution pattern of macroH2A.

Using peak calling to identify short to medium size peaks of macroH2A enrichment[51,59,60] we found that the frequency of macroH2A peaks was decreased by ~53% in LSH-deficient cells compared to wild-type controls and the length of the peaks reduced by ~12% (Fig. 5f). Moreover, the profile of macroH2A1 enrichment around previously identified macroH2A peaks[61] showed significant reduction of macroH2A at those sites in LSH-deficient MEFs compared to control MEFs (Fig. 5g). Analysis of diffuse macroH2A enrichment covering extended genomic regions[62,63] revealed that the number of broad macroH2A domains was reduced by ~28% in LSH-deficient cells compared to wild-type controls, and the length shortened by ~13% (Fig. 5h). Thus, the reduction of macroH2A deposition in the absence of LSH based on ChIP-seq analysis was in accord with the estimates based on chromatin association which suggested that macroH2A incorporation in chromatin is reduced ~40% in Lsh-deficient cells compared to wild-type controls (Fig. 4a and Supplementary Fig. 7a). It also indicated that factors other than LSH can promote macroH2A chromatin incorporation, since macroH2A domains were not completely absent in LSH-deficient cells. This notion was consistent with our earlier observations of combined LSH and macroH2A depletion (Figs. 3g, i and 4c, and Supplementary Figs. 5e, g and 7c, f), which showed that macroH2A depletion showed an additive effect in LSH-deficient cells.

Previous RNA-seq analysis had shown that only a modest subset of genes exhibited altered transcript steady state levels in LSH-deficient MEFs compared to wild-type MEFs and pathway analysis had suggested a role in the host response[30]. The subset of genes that are de-regulated (up or down) showed significantly greater differences of macroH2A occupancy in LSH-deficient cells compared to the average gene, while H2B enrichment showed no differences (Fig. 5i). Two thirds of macroH2A peaks overlap with repeat sequences (Supplementary Fig. 8d) and the number of peaks containing repeats is reduced by half in LSH-deficient cells (Fig. 5j). Repeat sequences show a 4-fold increase in expression level (Fig. 5k), suggesting an association between macroH2A reduction and repeat sequences expression in LSH-deficient cells.

Collectively, our data indicates global changes in macroH2A incorporation with reduced frequency and size of peaks, and broad domains associated with de-regulation of a subset of genes and repeat sequences in LSH deficient cells.

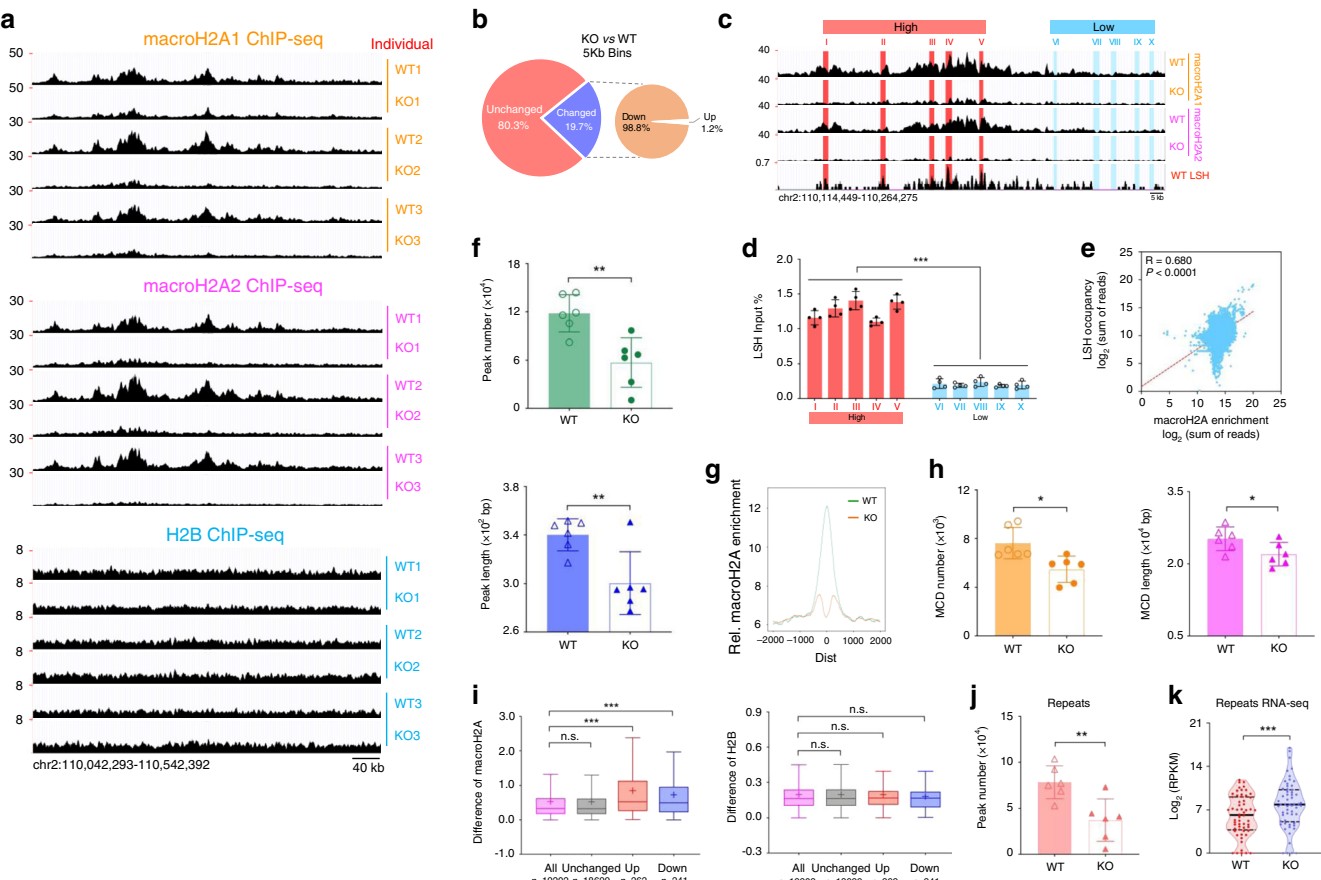

**Fig. 5 LSH depletion leads to genome-wide distribution changes of macroH2A enrichment. a** Genome browser view of macroH2A1, macroH2A2, and H2B nChIP-seq data representing three independent biologic replicates of *Lsh* WT and KO MEFs. **b** Illustration of genome-wide changes of macroH2A deposition in KO MEFs compared to WT cells. 19.7% of the genome shows altered ($p < 0.05$) macroH2A enrichment, 98.8% of these changes exhibit reduced macroH2A. **c, d** Representative genome browser snapshots of macroH2A1, macroH2A2, and LSH ChIP-seq data (**c**). ChIP-qPCR analysis validated LSH enrichment at the examined sites with high and low occupancy (**d**). ***$p < 0.0001$. **e** Scatterplot illustrating the correlation between macroH2A enrichment and LSH occupancy. The Pearson correlation coefficient is shown. **f** Frequency (**$p = 0.0030$) and length (**$p = 0.0072$) of macroH2A peaks in *Lsh* WT and KO MEFs. **g** Profile of macroH2A enrichment in KO MEFs (orange line) compared to WT MEFs (green line) around previously identified macroH2A peaks (+2Kb, −2Kb). **h** Frequency (*$p = 0.0108$) and length (*$p = 0.0476$) of broad macroH2A chromatin domain (MCD) in WT and KO MEFs. **i** Tukey-style boxplots representing the absolute value of macroH2A and H2B (WT minus KO) enrichment changes in subsets of de-regulated genes (up, $n = 262$; down, $n = 241$) and unchanged genes ($n = 18,699$) compared to all genes ($n = 19,202$). Cross dots (+), sample mean; center lines, median; boxes, 25–75 percentiles; whiskers, 1.5 IQR. Outliers are not shown. ***adjusted $p < 0.0001$, n.s. means not significant. **j** Frequency of macroH2A peaks at repeat sequences in WT and KO MEFs. **$p = 0.0017$. **k** Violin plot depicting relative RNA expression of repeat sequences in WT and KO MEFs. Endpoints depict minimum and maximum values, quartiles depicted by thin black lines, median depicted by thick black line. ***$p < 0.0001$. Data are represented as mean ± SD. Paired (**d, k**) and unpaired (**f, h, j**) two-tailed Student's *t* test; one-way ANOVA with Tukey's multiple comparison test (**i**). **d** representative of four independent experiments; (**f, h, j**) representative of six biologically independent samples. Source data are provided as a Source Data file.

**ICF4 mutations.** To understand the molecular underpinnings of ICF4 and the relation to macroH2A deposition, we introduced *LSH* disease-associated ICF4 mutations[17] into genetically modified ES cells. ICF4 mutations occur in regions that are highly conserved between human and mouse LSH (Fig. 6a). Three mutants were expressed in engineered ES cells at levels comparable to that of wild-type LSH (Fig. 6b). After addition of rapamycin, all ICF4 mutants were effectively tethered to the modified *Oct4* locus and showed a similar time kinetic compared to wild-type LSH (Fig. 6c). All ICF4 mutants showed reduced capacity in GFP repression (Fig. 6d). While in wild-type cells ~87% showed GFP repression upon rapamycin treatment, only 1% of cells expressing mutant S 745 Rfs*4 were capable to suppress GFP, only 12% of mutant Q 682R and 9% of mutant L 784 del cells (Fig. 6d). Remarkably, none of the ICF4 mutants was able to fully deposit macroH2A compared to wild-type LSH (Fig. 6e) supporting further the notion that repression is mediated via

macroH2A. We hypothesized that these mutants may impair LSH function in different ways and thus investigated their ability to interact with macroH2A (Fig. 6f). Wild-type LSH was used as positive control, as it showed ready interaction with macroH2A in a CO-IP assay. Interestingly, mutant Q 699R and mutant L 801 del, but not mutant S 762 Rfs*4 exhibited reduced efficiency to pull down the histone variant macroH2A (Fig. 6f) in U2OS cells thus suggesting that macroH2A deposition of these former ICF4 mutants was impaired based on their reduced ability to interact with macroH2A.

Finally, we examined a lymphocyte cell line derived from an ICF4 (L 801 del) patient[17] and compared it to a cell line derived from a parent (Fig. 7a). Western blot analysis revealed a slight loss of macroH2A chromatin incorporation in the ICF4 patient cell line compared to the parental cell line, while the total amount of macroH2A proteins were not changed (Fig. 7b). Using ChIP-qPCR analysis, we found a significant reduction of macroH2A

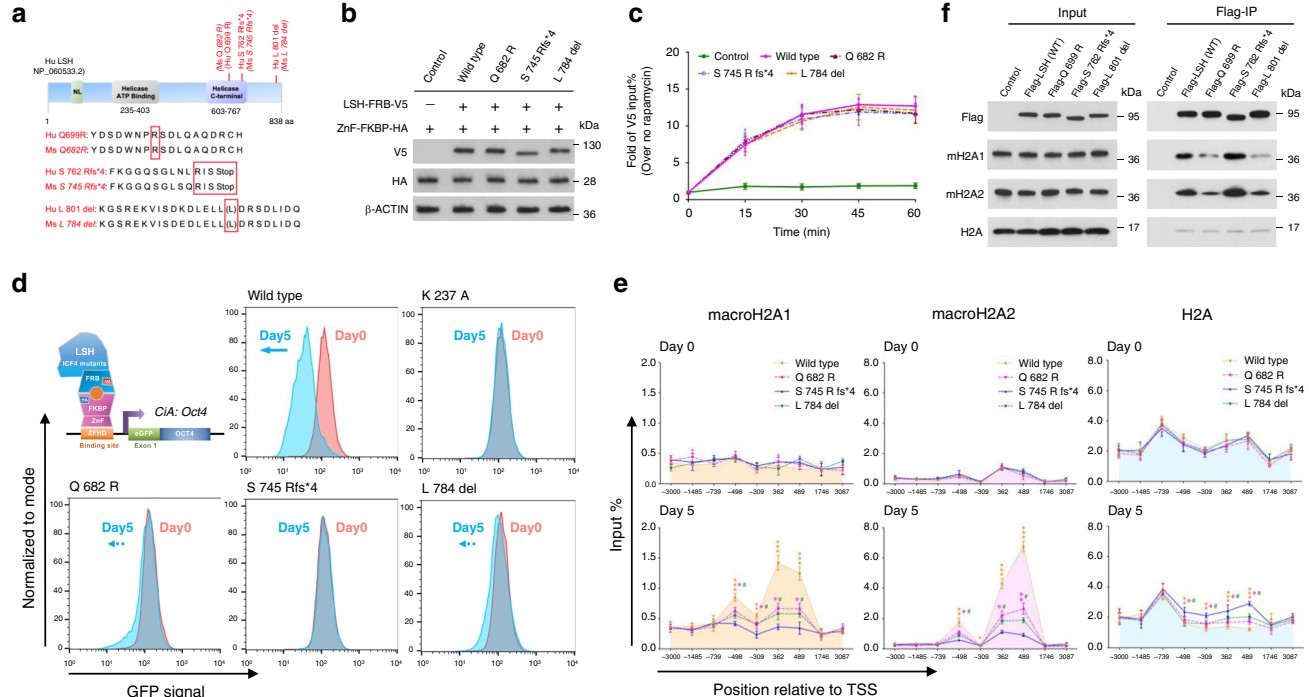

**Fig. 6 ICF4 derived LSH mutations fail to induce transcriptional repression and macroH2A deposition. a** Schematic representation of ICF4 associated human LSH mutations (red box) and alignment with murine LSH. **b** Western blot analysis to confirm the expression of three LSH-FRB-V5 mutant proteins and ZnF-FKBP-HA protein in ICF4 associated LSH CIP system compared to that in wild-type LSH CIP system using V5 and HA antibodies. Cells lacking LSH-FRB-V5 fusion protein expression served as controls. **c** ChIP-qPCR analysis to determine the recruitment of ICF4 associated LSH mutants and wild-type LSH to the engineered *Oct4* locus in the CIP system after rapamycin treatment at indicated time points using V5 antibody ($n = 3$ independent experiments). Cells lacking LSH fusion protein expression served as controls. Data are represented as mean ± SD. **d** Flow cytometry was used to measure reporter-GFP expression in ICF4 derived LSH mutants CIP system compared to that in wild-type or ATP mutant LSH (K 237 A) CIP system after 0 (red) and 5 (blue) days treatment of rapamycin. **e** ChIP-qPCR analysis to assess the dynamic changes of macroH2A1, macroH2A2 and H2A enrichment at the *CiA: Oct4* locus in the engineered mouse ES cells tethered with ICF4 associated LSH mutants or wild-type LSH with rapamycin treatment for 0 and 5 days ($n = 3$ independent experiments). Data are represented as mean ± SD. Significance assessed using one-way ANOVA with Tukey's multiple comparison test (wild type: *adjusted $p < 0.05$, ** adjusted $p < 0.01$ and ***adjusted $p < 0.001$; Q 682 R: ❖adjusted $p < 0.05$ and ❖❖adjusted $p < 0.01$; L 784 del: #adjusted $p < 0.05$). **f** Flag-IP of indicated proteins confirmed by western blot analysis in U2OS cells with a stable expression of wild-type Flag-LSH or ICF4 associated Flag-LSH mutants. Cells without a stably integrated of Flag-LSH expression vector served as controls. Source data are provided as a Source Data file.

deposition at specific loci. MacroH2A1 and macroH2A2 enrichment levels were significantly reduced at ribosomal genes (Fig. 7c, d). Furthermore, both types of macroH2A incorporation were significantly decreased at repeat sequences including Satellites and LINE1 elements compared to control parental cells (Fig. 7e). Correspondingly, transcripts levels of rDNA (18S and 28S) and repeats (SAT2 and LINE1) were increased in ICF4 patient cells (Fig. 7f, g) comparable to those we had observed after LSH knockdown or macroH2A knockdown. Collectively, ICF4 (L 801 del) mutant cells display impaired macroH2A deposition associated with transcriptional de-repression which may contribute to the pathophysiology of the syndrome.

## Discussion

A few factors have been found to remove macroH2A from its genomic location, but no deposition machinery has been identified which is involved in genome wide incorporation[9]. For example, ATRX inhibits macroH2A accumulation at telomeres and at the alpha globin locus, and FACT removes macroH2A from transcriptionally active regions[3,59]. Our study suggests that LSH promotes incorporation of macroH2A into chromatin for part of the genome. We show here that tethered LSH induces macroH2A1.2 and macroH2A2 accumulation at the recruitment site and that H2A is reciprocally reduced. In addition, the ATP binding site of LSH is required for macroH2A1.2 and

macroH2A2 deposition implying chromatin remodeling activity of LSH in the process. LSH can interact with macroH2A, and this interaction shows some specificity, since two ICF4 mutant proteins and the histone variant H2A.Z do not show the same degree of interaction. Furthermore, under physiologic conditions, LSH is critical for the genome wide establishment of broad macroH2A domains including heterochromatin containing repeat sequences and ribosomal genes. Thus, LSH is either directly involved as a histone exchange factor or indirectly, coordinating the exchange reaction.

Chromatin remodeling enzymes of the SNF2/Rad54 helicase family, to which LSH belongs, are evolutionarily conserved and can participate in chromatin assembly, repositioning of nucleosomes or exchange of histone variants[24,25,64]. Most chromatin remodelers can use their ATP-dependent DNA translocase activity to slide nucleosomes along DNA, but histone replacement can also occur without repositioning of nucleosomes[3,24]. Some factors such as INO80C can slide nucleosomes, as well as perform histone exchange. Notably, LSH and its binding partner CDCA7 can increase nucleosome mobility in vitro in a restriction enzyme accessibility assay[26]. Interestingly, we observed an asymmetrical deposition of macroH2A1.2 and macroH2A2 with respect to LSH binding. Most remodelers anchor at a fixed position on the histone octamer and show unidirectional sliding along DNA[4,24]. The recruitment of LSH at the genetically modified *Oct4* locus may give LSH a certain orientation and the translocase domain of LSH

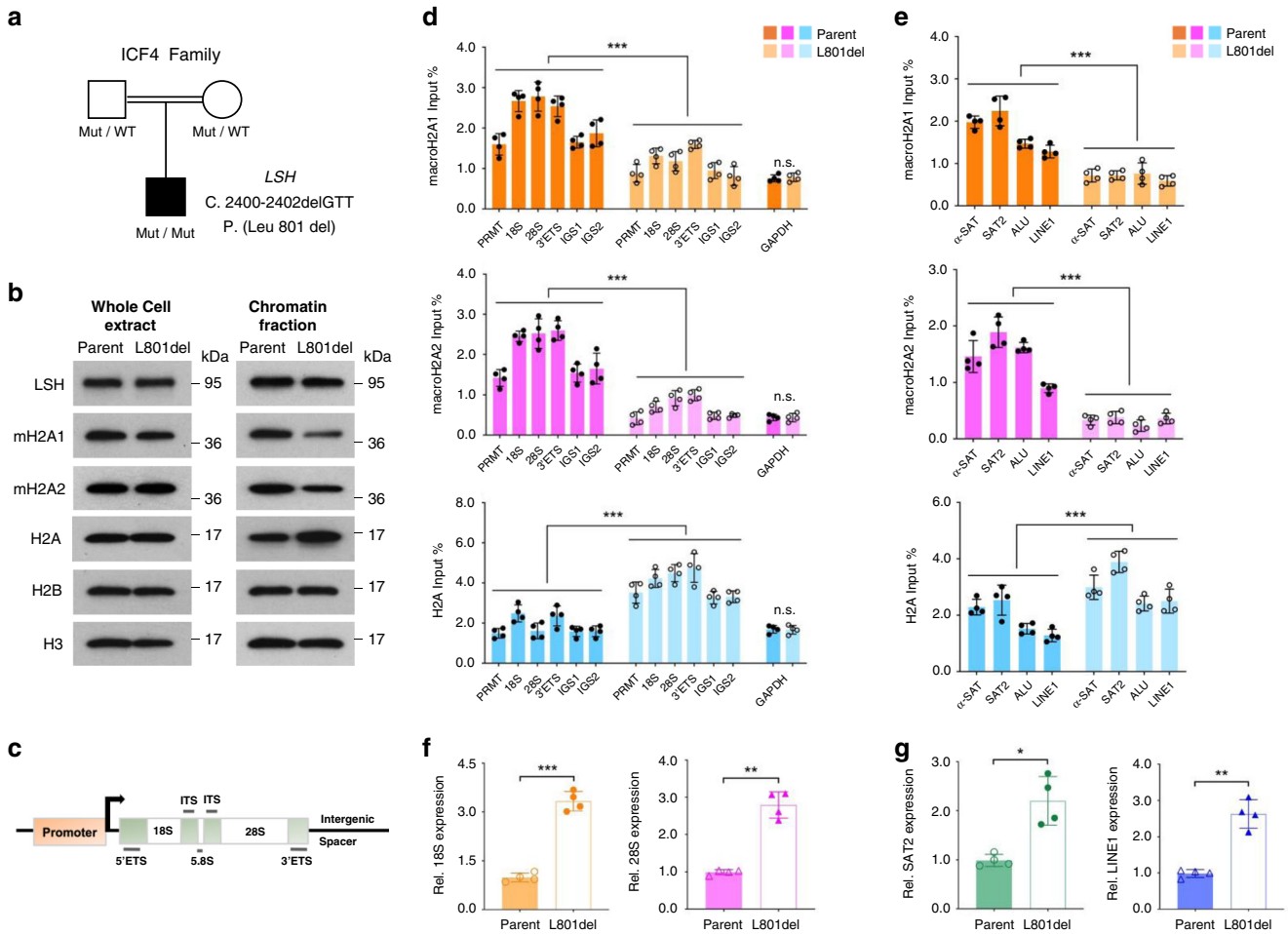

**Fig. 7 Human ICF4 mutant cells display impaired macroH2A deposition associated with transcriptional de-repression at repeats. a** Schematic representation depicts a human homozygous Leu 801 del (L 801 del) LSH mutant ICF4 patient family tree. Both parents are heterozygous for the mutated allele. **b** Western blot analysis for detection of indicated proteins in whole-cell extract or chromatin fraction isolated from ICF4 (L 801 del) patient and parental (Parent) lymphocyte cells. **c, d** Schematic representation of human rDNA repeated locus (**c**). PRMT promoter, ETS external transcribed spacer, ITS internal transcribed spacer, IGS intergenic spacer. ChIP-qPCR analysis shown in **d** for detection of macroH2A1, macroH2A2 and H2A enrichment at rDNA sequences in ICF4 patient cells compared to parental control cells. ***$p < 0.0001$. **e** ChIP-qPCR analysis for detection of macroH2A1, macroH2A2 and H2A enrichment at repeat sequences (satellite sequences, ALU and LINE1 elements) in ICF4 (L 801 del) and parental cells. ***$p < 0.0001$. **f, g** RT-qPCR analysis for detection of transcriptional levels at rDNA (18S, ***$p = 0.0003$; 28S, **$p = 0.0051$; **f**) and repeats (SAT2, *$p = 0.0104$; LINE1, **$p = 0.0048$; **g**) in ICF4 mutant cells compared to parental lymphocyte cells. Data are represented as mean ± SD. Paired two-tailed Student's $t$ test (**d–g**). **d–g** representative of four independent experiments. Source data are provided as a Source Data file.

may impose unidirectional movement along DNA, which facilitates H2A-H2B removal and subsequent marcoH2A-H2B deposition. Known chromatin remodelers that act as nucleosome editors include SWR1 and p400 that replace H2A-H2B dimers with H2A.Z, INO80C which performs the reverse reaction or can remove the histone variant H2A.X and p400 that replaces H3.1 with the variant H3.3[3,24]. Our data demonstrates that LSH is a critical factor in genome wide macroH2A deposition. However, several observations suggest that LSH is not the only factor and redundancy may exist. For example, we observed ~30–50% reduction in broad domains and peaks consistent with the ~40% decrease in chromatin association comparing LSH deficient cells to wild-type controls. This indicates a widespread effect of LSH on genomic macroH2A distribution, however, some macroH2A was still deposited in the absence of LSH. In addition, the combination of macroH2A and LSH deletion revealed that macroH2A deletion had an additive effect on LSH deletion in terms of transcription and nuclear size, while LSH deletion did not exacerbate macroH2A deletion. This suggests that, LSH may mediate most of its effect on transcription and nuclear size via

macroH2A, but that other so far unknown factors may also be capable to promote macroH2A deposition or may be able to substitute for LSH.

A recent report has shown that transcription is an important mechanism to remove macroH2A after it has been deposited across the genome by yet unknown factor(s)[12]. However, the pruning mechanism alone may not explain why low levels of macoH2A are found at gene-poor sites[61] and how macroH2A enrichment can be decreased in LSH-deficient cells at sites that do not show any transcriptional changes. Thus, our model suggests that multiple pathways, including a LSH dependent pathway and the transcriptional pruning pathway via the histone chaperone FACT[12], act together to shape genome wide macroH2A deposition. Our model proposes that LSH deposits macroH2A at specific regions of the genome, and transcription prunes and fine tunes the amount of macroH2A at any given transcriptionally active genomic site.

If some effects of LSH were mediated via macroH2A deposition, we would expect that LSH and macroH2A depletion show a likeness in their phenotypes. There is some degree of similarity,

for example, macroH2A or LSH depletion reduces polycomb repression at *Hox* cluster target genes[65,66]; removal of macroH2A increases the efficiency of induced pluripotency[67] and LSH removal enhances cellular plasticity[68]; knockdown of macroH2A or LSH in mouse ES cells results in incomplete silencing of pluripotency genes *Oct4*, *Nanog*, and *Sox2* after differentiation[45,69,70]; knockout of macroH2A or LSH in mice results in smaller body size of day18.5 fetus and mice show increased perinatal lethality (30% and 100% respectively), but neither phenotype shows a pathology that would point to the cause of death[53,71]. Protein coding genes display only modest deregulation in either macroH2A or LSH-deficient cells, however, diverse repeat elements and retroviral elements are transcriptionally derepressed[10,30,50,53]. There are reproductive problems in macroH2A knockout mice in C57BL/6 background and *Lsh* knockout mice show defects in male and female germ cell development[18,53,72]. While macroH2A knockout mice share some phenotypic similarities with *Lsh* knockout mice, not all LSH effects maybe mediated by macroH2A. For example, *Lsh* knockout mice exhibit more severe lethality and LSH-deficient cells still have a remainder of macroH2A deposition, suggesting that they do not share a completely congruent pathway.

Importantly, we also found that ICF4 mutations fail to incorporate macroH2A at tethered sites and show impaired capacity for transcriptional silencing. Histone exchange is a multi-step process[24,73] and the nature of the mutants' malfunction maybe manifold. We observed reduced interaction of two ICF4 mutants with macroH2A which may explain their impaired ability to deposit macroH2A. Other potential malfunctions of ICF4 mutant proteins may include the ability to perform DNA translocation, to mobilize the H2A-H2B dimers or to interact with other factors of a presumed LSH complex that may be critical for macroH2A-H2B dimer recognition and the exchange reaction[24,73]. ICF4 (L 801 del) patient cells lack normal macroH2A accumulation at several repeat sequences and fail to repress those loci, indicating that ICF4 patients suffer from abnormal histone variant deposition. To our knowledge, macroH2A knockout mice have not yet been assessed for some of the hallmarks of the ICF4 syndrome such as immunoglobulin deficiency, genomic instability and neurologic malfunction. Several genetic diseases have been uncovered, involving mutations in a chromatin remodeler, chaperone, or histone variant, but no disease has been as yet attributed to abnormal macroH2A deposition[44,74,75]. MacroH2A as well as LSH has been implicated in processes of transcriptional regulation, DNA repair, genomic stability, and tumor development[9,43,56,76–78] and it will be important in the future to discern which molecular pathways partially or completely overlap and contribute to the pathophysiology of the ICF4 syndrome.

## Methods

**Cell culture**. The *CiA: Oct4* mouse ES cell line[34] contains one modified *Oct4* allele harboring one array of DNA binding sites (12 × ZFHD1) in the promoter region upstream of an in-frame nuclear eGFP reporter inserted at the ATG of exon 1 of *Oct4*. WT (*Lsh+/+*) and KO (*Lsh−/−*) mouse ES cell lines had been derived from mouse embryos, and KO ES cells had been transfected with *Lsh* WT or *Lsh* mutant (K 237A) expression vector[27]. All mouse ES cells were grown on 0.1% gelatin (Sigma) coated six-well plates in KnockOut™ DMEM (Gibco) media supplemented with 15% KnockOut™ Serum Replacement (Gibco), 10 mM HEPES pH 7.5 (Gibco), 1× Minimum Essential Media (MEM) nonessential amino acids (Gibco), 1× GlutaMAX™ (Gibco), 50 U ml−1 Pen/Strep (Gibco), 0.1 mM 2-mercaptoethanol (Sigma), and 1:10,000 ESGRO® Recombinant Mouse LIF Protein (Millipore). WT (*Lsh+/+*) and KO (*Lsh−/−*) MEF cell lines[30] were grown in high-glucose DMEM (Invitrogen) supplemented with 10% Fetal Bovine Serum (FBS, Omega Scientific), 1× GlutaMAX™ and 50 U ml−1 Pen/Strep. U2OS (ATCC), HEK293T (ATCC), and normal human lung fibroblast (Coriell Institute for Medical Research) cell lines were cultured in high-glucose DMEM (Invitrogen) supplemented with 10% FBS and 50 U ml−1 Pen/Strep. ICF4 parent and patient cell lines (gift from Dr. Claire Francastel and Dr. Guillaume Velasco,

Paris, France)[17] were grown in RPMI-1640 (Gibco) media supplemented with 20% heat-inactivated FBS (Omega Scientific), 1× GlutaMAX™ and 50 U ml−1 Pen/Strep. All cells were cultured at 37 °C in a humidified incubator containing 5% $CO_2$.

**Cell treatment**. Lentivirus was produced by Lipofectamine 2000 (Invitrogen) transfection of HEK293T cells with gene delivery vector co-transfected with packaging vectors pspax2 (Addgene) and pMD2.G (Addgene)[79]. Concentrated lentiviral supernatants by Lenti-X™ Concentrator (Takara) were used to infect *CiA: Oct4* mouse ES cells with 8 μg ml−1 polybrene (Sigma). Cells were incubated overnight prior to virus removal and selection of lentiviral construct achieved in *CiA*: Oct4 with either: ZnF-FKBP-HA plasmid, puromycin (2 μg ml−1) or LSH (WT or Mutant)-FRB-V5 plasmid, hygromycin (400 μg ml−1). ZnF-FKBP-HA fusion protein was directly recruited to the indicated DNA binding domains (12 × ZFHD1) and used to harbor LSH (WT or Mutant)-FRB-V5 fusion protein. Proximity of FRB and FKBP was induced by addition of rapamycin (Selleckchem) at 3 nM in all experiments. *Lsh* MISSION shRNA-expressing lentiviral vector was from Sigma. After lentivirus production and infection, U2OS cells were selected with puromycin (2 μg ml−1) to generate stable cell line. Transient transfections of U2OS cells with 3 × FLAG-LSH (WT or ICF4-Mutant)-IRES2-GFP, macroH2A2-GFP, and GFP vectors were performed by using TurboFect transfection reagent (Invitrogen). Afterwards, two continuous rounds of fluorescence-activated cell sorting (FACS) were utilized to collect respective GFP positive U2OS cells in 3 and 15 days in order to generate stable cell line. macroH2A1 and macroH2A2 siRNAs (ON-TARGETplus SMARTPool, Dharmacon) were transfected using Lipofectamine RNAiMAX transfection reagent (Invitrogen) following the manufacturer's instructions and analyzed 48–72 h post transfection.

**Flow cytometry analysis**. All flow cytometry analyses were performed on a LSR II (BD Biosciences) and analyzed with FlowJo v. 10 software. Cell population of each sample was gated using unmodified mouse ES cells without GFP expression as negative controls. Individual cells were gated based on forward and side scatter, autofluorescent cells were omitted, and remaining cells were analyzed for GFP level.

**RT-qPCR analysis**. Total RNA was isolated using Rneasy plus Mini kit (Qiagen) following with additional RNase-free DNase treatment. Reverse transcription was performed by using PrimeScript RT Reagent Kit (Takara). Transcripts were amplified using MyiQ2 two colors Real-Time PCR detection system (Bio-rad) with iTaq Universal qPCR Mastermix (Bio-rad). Relative expression was normalized to internal *Gapdh* abundance. Primers are listed in Supplementary Table 1.

**ChIP-qPCR assay**. For ChIP assays[80] 1 × 10^7 cells were cross-linked with 1% formaldehyde, lysed, and sonicated on ice to generate 200–800 bp DNA fragment. In all, 1% of each sample was saved as input fraction. Immunoprecipitation was performed using chip-grade antibodies listed in Supplementary Table 3. After reversal of cross-linking, precipitated DNA was suspended in 50 μl of Nuclease-Free water and analyzed by qPCR. The normalization method for ChIP analysis is percent of input. The specific ChIP primers are shown in Supplementary Table 1.

**Bisulfite sequencing analysis of DNA methylation**. Genomic DNA was extracted using the DNeasy blood & tissue kit (Qiagen). For each reaction, 1 μg genomic DNA was bisulfite converted with the EpiTec Bisulfite Kit (Qiagen). Knock-in-specific eGFP sequence was amplified by PCR. The PCR products were cloned by using TOPO TA cloning kit (Invitrogen). IAP repetitive element was used as a positive control. After screening of positive clones and digestion of plasmids with EcoRI (NEB), 10 clones for each sample were sequenced to identify the methylation level of cytosine. Methylation profiles were analyzed by using BiQ Analyzer version 0.7 software[81]. Primers for PCR amplification are listed in Supplementary Table 1.

**Chromatin isolation and immunoblotting**. For cell chromatin fractionation[82], buffers were supplemented with 0.5 mM dithiothreitol (DTT), 0.1 mM phenylmethyl sulfonyl fluoride (PMSF) and 1× protease inhibitor cocktail. About 1 × 10^7 cells were resuspend cells in 1 ml buffer A (0.1% Triton X-100, 10 mM HEPES pH 7.9, 10 mM KCl, 1.5 mM MgCl2, 0.34 M sucrose, and 10% glycerol) and incubated on ice for 10 min. Nuclei were isolated by centrifugation at 1500×*g*, 4 °C. The supernatant was taken as cytosolic fraction. Nuclei were washed once with buffer A and lysed for 30 min in buffer B (3 mM EDTA and 0.2 mM EGTA) on ice. Chromatin fraction was pelleted by centrifugation at 1500×*g*, 4 °C. Whole-cell extract were prepared using 2 × 10^6 cells. Cells were lysed on ice for 15 min with 150 μl RIPA buffer (25 mM Tris-HCl pH 7.6, 150 mM NaCl, 0.1% SDS, 1% sodium deoxycholate, and 1% NP-40). The lysate was sonicated for 30 s with 50% pulse and centrifuged for 15 min at 14,000×*g*, 4 °C. The supernatant was collected, and protein concentration was measured by bicinchoninic acid assay (BCA) assay (Thermo Fisher). Samples were boiled in Laemmli sample buffer (Bio-Rad) at 95 °C for 5 min. Equal amounts of protein were loaded onto acrylamide/bis gels and transferred to PVDF membranes after electrophoresis. Following blocking in 5% nonfat milk for 1 h, membranes were incubated at

4 °C overnight in primary antibodies listed in Supplementary Table 3. After incubation with HRP-conjugated secondary antibodies as listed in Supplementary Table 3 at a dilution of 1:2500 for 1 h, Amersham ECL western blotting analysis system was used for signal detection.

**Co-immunoprecipitation analysis.** For CO-IP analysis $2 \times 10^7$ U2OS cells expressing 3× FLAG-LSH (WT or ICF4-Mutant) and macroH2A2-GFP were collected and washed twice with PBS. Cell pellets were resuspended with IP lysis buffer (25 mM Tris-HCl pH 7.4, 150 mM NaCl, 1% NP-40, 1 mM EDTA, and 5% glycerol) supplemented with fresh 0.5 mM DTT, 0.1 mM PMSF, and 1× protease inhibitor cocktail. Incubate cell lysates on ice for 10 min with periodic mixing. Remove cell debris by centrifugation at 13,000×g for 10 min at 4 °C. The supernatant was collected and protein concentration was measured by BCA assay. For each IP, 1 mg protein was diluted with IP lysis buffer to 1 ml, and 50 µl was taken as input. Precleaning was performed by addition of 50 µl Protein G Magnetic Beads (Bio-rad) and incubation for 1 h rotation at 4 °C. Each precleaned sample was mixed with 4 µg antibody respectively listed in Supplementary Table 3 followed by rotation at 4 °C overnight. Afterwards, 50 µl magnetic beads were added to each sample with 2 h rotation at 4 °C. After washing the beads with IP lysis buffer four times and with PBS twice, immunoprecipitated proteins were eluted by addition of 1× Laemmli sample buffer and incubation at 95 °C for 5 min. IP samples were subsequently loaded onto acrylamide/bis gels and subjected to western blotting as described above. Primary antibodies used for immunoblotting were listed in Supplementary Table 3. As a control, cell lysates from U2OS cells expressing GFP were prepared and performed the same assay.

**Immunofluorescence staining and nuclei area measurement.** Cells grown on chamber slides were fixed in 4% paraformaldehyde for 20 min, permeabilized with 0.1% Triton X-100 for 15 min, blocked in 5% BSA for 30 min, then incubated at 4 °C overnight in primary antibodies listed in Supplementary Table 3. The slides were subsequently incubated with Alexa fluorophore–conjugated secondary antibodies for 1 h as listed in Supplementary Table 3 at a dilution of 1:500. Finally, the slides were stained with DAPI and imaged by confocal microscopy. Area of nuclei was measured by staining with LAMIN B1 and analyzed with ImageJ v. 2 software.

**nChIP-seq.** For nChIP[83] all buffers were freshly supplemented with 0.5 mM DTT, 0.1 mM PMSF, and 1× protease inhibitor cocktail. Briefly, $3 \times 10^7$ cells were used for nuclei isolation. Cells were lysed with Buffer A (0.32 M sucrose, 0.2% NP-40, 15 mM Tris pH 7.5, 60 mM KCl, 15 mM NaCl, 5 mM MgCl2, and 0.1 mM EGTA) and incubated on ice for 10 min. The lysate was layered onto Buffer B (1.2 M sucrose, 15 mM Tris pH 7.5, 60 mM KCl, 15 mM NaCl, 5 mM MgCl2, and 0.1 mM EGTA). Nuclei were pelleted at 10,000×g for 20 min and gently resuspended in Buffer C (0.32 M sucrose, 50 mM Tris pH 7.5, 4 mM MgCl2, and 1 mM CaCl2). CaCl2 was added to 3 mM, 8.5 units of MNase (Thermo Fisher) was added and the mixture was incubated at 37 °C for 10 min. The reaction was stopped by adding EGTA to 10 mM and incubated on ice for 5 min. Nuclei were collected by centrifugation at 10,000×g for 7 min and supernatant was collected as S1. The pellet was gently resuspended in Buffer D (50 mM Tris pH 7.5, 300 mM NaCl, 2 mM EDTA, and 0.1% NP-40) and incubated for 2 h with head-to-head rotation at 4 °C. Nuclei were spun down at 10,000×g for 7 min and supernatant was collected as S2. S1 and S2 were pooled and further cleared at maximum speed for 5 min. Chromatin concentration was quantified spectroscopically (absorbance A260). For each immunoprecipitation, 100 µg chromatin was diluted with Buffer E (50 mM Tris pH 7.5, 150 mM NaCl, 2 mM EDTA, and 0.05% NP-40) to 1 ml, and 1% from each sample was taken as input. For each reaction, 3 µg antibody listed in Supplementary Table 3 was added and incubated at 4 °C overnight. In all, 30 µl Magna ChIP Protein A + G magnetic beads (Millipore) were added and incubated for 2 h. Beads were then washed once with Buffer G 150 (50 mM Tris pH 7.5, 150 mM NaCl and 0.5% NP-40), twice with Buffer G 250 (50 mM Tris pH 7.5, 250 mM NaCl, and 0.5% NP-40) and once with Tris-EDTA buffer (10 mM Tris pH 7.5 and 1 mM EDTA). Input and beads were incubated with 50 µg ml⁻¹ RNase A for 1 h at 37 °C in Tris-EDTA buffer. Samples were incubated in 0.5% SDS and 500 µg ml⁻¹ Proteinase K with constant mixing at 56 °C overnight. Supernatant was collected from the beads. Input and nChIP DNA were purified with MinElute PCR purification kit (Qiagen) and analyzed with Agilent high sensitivity DNA chip using Agilent Technologies 2100 Bioanalyzer. Sequencing libraries were constructed from DNA samples including input and nChIP DNA with the Illumina TruSeq V3 library construction protocol (Illumina). Sequencing runs were performed on an Illumina NextSeq 500 in 75-base-pair (bp) single-end mode at the CCR-Sequencing Facility, National Cancer Institute, Frederick. RTA 1.8.70.0 software was used for basecall analysis. Reads were trimmed using Cutadapt v.1.18 to remove adapters and low-quality flanking regions. These reads were aligned to the mouse genome (MM10) using Bowtie2 2.2.6. with a mismatch error rates ≤0.05%. Reads that mapped to multiple locations in the genome were discarded.

**nChIP seq analysis.** Genome browser views illustrate macroH2A and H2B enrichment for individual nChIP-seq samples expressed as read depth normalized to individual input controls, for example, KO1 represents macroH2A1 nChIP-seq sample of KO1 MEFs normalized to input control of KO1 MEFs. Supplementary Table 2 lists nChIP samples and corresponding input samples, and the number of read alignments per sample. LSH ChIP-seq results of WT MEFs were retrieved from the Gene Expression Omnibus database under accession number GSM835828. nChIP samples and their corresponding input samples were run through Macs v. 2.2.6[60] using the following workflow:

 macs2 callpeak -n $[sample]–bdg–nomodel–extsize 150 -t $[sample].input -c $[sample]

 macs2 bdgcmp -t $[sample]_treat_pileup.bdg -c $[sample]_control_lambda.bdg -m qpois -o $[sample]_qvalue.bdg

 bg2wig $[sample]_treat_pileup.bdg $[sample].wig mm10.chrom.sizes

 wigToBigWig -clip [sample].wig mm10.chrom.sizes [sample].bw

The outputs from call peak and bdgcomp created the pileup bedgraph (bdg) which was converted to a wig file using the custom bg2wig program. The pileups represent sequencing depth and background (input) normalized relative enrichment signal. Wigs were converted to bigwigs using the wigTotBigWig 4 program from UCSC[84].The bigwigs (*.bw files) were viewed using the UCSC browser[85]. To tabulate macroH2A enrichment or LSH enrichment in 5 Kb tiles across the genome, wig files were converted to 5 Kb bins using a custom program. Resulting bins were the sum of all read coverage in the 5-Kb regions. To determine genome wide changes of macroH2A deposition in KO MEFs compared to controls, we compared macroH2A enrichment of six KO samples to macroH2A enrichment of six WT samples for individual 5Kb bins across the genome (for autosomes). Each bin was interrogated for statistically significant differences of KO/WT ratios using a two-tailed t-test. The percentage of differentially enriched bins is presented in Fig. 5b. Pearson correlation for macroH2A and Lsh occupancy was based on the sum of read coverage in 5Kb tiles and computed using Excel Microsoft 365 ProPlus. To determine broad macroH2A domains, each sample reads were quality trimmed using Cutadapt v.1.18 with a minimum quality of 10 and minimum length of 10. Trimmed reads were aligned using bwa-mem with hard clipping on. The bwa-mem output was sorted and indexing using Samtools 1.10. The resulting bam files were run through Epic2 v. 0.0.41 to create peak call in bed files[62,63]. Peaks were selected with an FDR < 0.01, length >10 Kb, and $\log_2$ (Fold Change) >0.8. To compute macroH2A enrichment at protein coding genes, coverage of macroH2A enrichment at individual MM10 gene locations was extracted for each sample wig file using bigWigSummary[84]. Changes of macroH2A enrichment were computed for the gene body of protein coding genes ($n = 19,202$) using the z-test for statistical analysis. The differences are computed by subtracting the mean value of macroH2A enrichment for Lsh knockout samples (KO1, KO1RS, KO2RS, KO6, KO7, and KO8) for each gene from the mean value of corresponding wild-type samples (WT1, WT1RS, WT2RS, WT6, WT7, and WT8). Upregulated genes (up in Lsh knockout versus wild type, $n = 262$) and downregulated genes ($n = 241$) have been previously identified by RNA-seq analysis conducted in Lsh−/− and Lsh+/+ MEF samples[30]. Peaks were called using Macs v. 2.2.6[60]. For example, macs2 callpeak -n $[sample]–bdg–nomodel–extsize 150 -t $[sample].input -c $[sample]

Peak overlap with repeat regions was computed using the rodent repeat database rodrep_ref.txt file from RepeatMasker version 4.1.0 (http://www.repeatmasker.org/). We aligned rodrep_ref.txt against MM10 genome using the Blat program from UCSC toolkit v. 367[86]. Best results were filtered using the following code:

 pslReps -nearTop=0.02 -minCover=0.60 -minAli=0.85 -noIntrons repeats. mm10.psl repeats.best.psl out.psr > pslReps.out 2>&1

 psl2bed < repeats.best.psl > best.repeats.bed

Each repeats type was intersected with peaks using Bedops[87] program version 2.4.39 "intersect" option and counts were reported. The profile of macroH2A enrichment in KO1 MEFs compared to WT1 MEFs was computed for macroH2A2 peaks ($n = 3,305$) that had been previously identified in embryonic stem cells[61].

**Statistics and reproducibility.** All ChIP-qPCR and RT-qPCR experiments were repeated at least three times as independent biological replicates and results are presented as mean ± SD. ChIP-seq samples as well as their respective input samples were derived as biologic replicates from three murine embryonic fibroblast cell lines for each genotype. Each cell line was derived from an individual embryo. All immunoblots were repeated at least three times with different biological samples and led to similar results. Micrographs are representative of a minimum of six images taken from at least three biological replicates. For Figs. 1a, b, d, e; 3a, e, f, h; 4a, d, g, h; 6b, d, f; and 7b and Supplementary Figs. 1; 3d; 4a; 5a, f; and 7a, d, at least three independent biological samples were performed with similar results and representative results are shown. Relative RNA expression of repeat sequences (Fig. 5k) computed from RNA-seq data was calculated as the value of $\log_2$ (RPKM + 1). MacroH2A and LSH occupancy levels (divided in fixed 5 Kb tiles) were transformed as the value of $\log_2$ (sum of reads + 0.1) to calculate Pearson correlation coefficient (Fig. 5e and Supplementary Fig. 8a), and to statistically analyze macroH2A and LSH occupancy levels at the macroH2A changed and unchanged sites (Supplementary Fig. 8b, c). Statistical analyses were performed with GraphPad Prism v. 8 software using paired or unpaired two-tailed Student's t test between two samples and one-way ANOVA with Tukey's analysis for multiple comparisons unless otherwise stated. $P < 0.05$ was considered a statistically significant difference.

**Reporting summary**. Further information on research design is available in the Nature Research Reporting Summary linked to this article.

## Data availability

RNA-seq data was retrieved from our recently published work and is available at GEO database (GSM1382355, GSM1382356, GSM1382357, and GSM1382358)[30]. LSH ChIP-seq data in WT MEFs was retrieved from GEO database (GSM835828)[58]. The macroH2A and H2B ChIP-seq data has been deposited in Gene Expression Omnibus, GEO accession number: "GSE142082 [https://www.ncbi.nlm.nih.gov/geo/query/acc.cgi?acc=GSE142082]". All other data supporting the findings of this study are available within the paper and its Supplementary Information files and are available from the corresponding author upon reasonable request. A reporting summary for this article is available as a Supplementary Information file. Source data are provided with this paper.

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

## Acknowledgements

We wish to thank Karen Saylor from the FNLCR LASP Animal Research and Technical Support Staff and members of the NCI-Frederick Flow Cytometry Core and NCI-Frederick Sequencing Core. We thank Dr. Claire Francastel and Dr. Guillaume Velasco for the generous gift of ICF4 cells. This project has been funded in whole or in part with federal funds from the National Cancer Institute, National Institutes of Health, under contract HHSN26120080001E. The content of this publication does not necessarily reflect the views or policies of the Department of Health and Human Services, nor does mention of trade names, commercial products, or organizations imply endorsement by the U.S. Government. This Research was supported by the Intramural Research Program of the NIH, National Cancer Institute, Center for Cancer Research. Frederick National Laboratory for Cancer Research is accredited by AAALAC International and follows the Public Health Service Policy for the Care and Use of Laboratory Animals. Animal care was provided in accordance with the procedures outlined in the "Guide for Care and Use of Laboratory Animals" (National Research Council; 1996; National Academy Press; Washington, D.C.).

## Author contributions

K.N. and K.M. conceived and designed experiments. K.N., J.R., X.X., and Y.H. conducted experiments. N.A.H., S.M.G.B., and G.R.C. provided crucial reagents, expertize, and guidance. R.F. performed bioinformatic analysis. K.N. and K.M. analyzed data and wrote the manuscript.

## Competing interests

The authors declare no competing interests.
