## [Peer Review File · Nature Communications]

Reviewers' Comments:

Reviewer #2:

Remarks to the Author:

In this manuscript Ni et al. establish that the LSH chromatin remodeler mediates gene repression, in part, through inducing the deposition of the histone variant macroH2A. In Figure 1, the authors use a chemical genetic approach to induce the recruitment of LSH to a GFP:OCT4 gene report by rapamycin treatment in ES cells and show that LSH recruitment represses the gene about 2 fold over a 5 day period in a manner that requires LSH's ATPase activity. In figure 2 the authors show that macroH2A1 and macroH2A2 recruitment occupancy to the Oct4 reporter increases in a LSH ATPase dependent manner over a 3 day rapamycin treatment. Next, the authors show that Lsh promotes macroH2A1 recruitment to rRNA genes and a variety of other repeat sequences (Fig 3). In figure 4, the authors argue that Lsh loss leads to a global reduction in macroH2A1 deposition using chromatin fractionation and nuclear size as a surrogate. In figure 5, the authors use ChIP-seq to demonstrate loss of macroH2A deposition genome wide. In figure 6, the authors examine the role of an LSH mutation found in a genetic disease, immunodeficiency centromeric instability facial anomalies (ICF), and provide evidence that this mutation affects macroH2A deposition. Finally, in figure 7 demonstrate that human cells from a patient with ICF4 have a reduction in macroH2A1 deposition at repeats.

Major comments:

1. This is an interesting manuscript that makes an important discovery regarding the role of Lsh in mediating macroH2A deposition for the purposes of silencing. Identifying the machinery responsible for macroH2A deposition has been a roadblock for the field and so this paper represents an advance that will be of great interest. However, there are a variety of issues that the authors need to address before the manuscript is suitable for publication
2. The manuscript implies that Lsh is a chromatin remodeler capable of depositing macroH2A1 into chromatin, in a manner analogous to how the SWR1 complex mediates the deposition of H2AZ. Another possibility is that Lsh acts upstream of macroH2A deposition through its chromatin remodeling function but that some other factor mediates macroH2A deposition into chromatin. If Lsh is directly responsible for macroH2A deposition, the author should show that Lsh interacts with macroH2A and is selective for macroH2A over other H2A-type histones.
3. Again if Lsh is directly mediating macroH2A deposition, ChIP-seq should be able to show that Lsh and macroH2A localize to the same genomic regions. No Lsh ChIP is presented.
4. The model presented by the authors is in stark contrast to the kinetic model presented by the Bernstein lab where macroH2A is shown to be deposited everywhere and is specifically removed from locations. Discussion on how/if these two models can coexist needs to be presented.
5. The authors make the point that the Lsh KO in mice recapitulates many features of ICF, but to my knowledge macroH2A KOs do not recapitulate features of ICF. This needs to be discussed.
6. MacroH2A is not only associated with transcriptional silencing and not only associated with constitutive heterochromatin (as expressed on line 62). It is also found in facultative heterochromatin and euchromatin. MacroH2A is a family of variants, macroH2A1.1, macroH2A1.2 and macroH2A2. MacroH2A1.1 is distinct in both function (it plays a positive role in transcription) and localization (It resides in euchromatic regions). Which macroH2A1 variants are expressed in these cells should be quantified (typically ES cells do not express much macroH2A1.1). If so, this should be clarified so as not to confuse the field about the function of these distinct histone variants.
7. Fig 1: There is a disconnect between where LSH is recruited and where macroH2A1 deposition occurs. I can see how macroH2A might not be able to be recruited directly at the site of LSH recruitment, but why would it be asymmetrical – macroH2A1 is only getting recruited to the gene body, not a similar distance upstream? This needs to be discussed.
8. There are no statistics in any of the critical ChIPs demonstrating the significance of change in factor recruitment (Fig 2, Fig 6E).
9. In Fig 4, the authors suggest that there is less chromatin associated macroH2A while total protein levels remain the same in the Lsh KO. But it's not much reduced in chromatin, so where is

all of that macroH2A going and how is that result consistent with the macroH2A ChIP-seq data shown in Fig 5?

Minor comments:

10. Fig 1C and 6C : ChIP should be expressed as % input rather than fold enrichment (which is ambiguous) as done in 1F
- 11.
12. Fig 2F: What is the positive control to suggest the H1.2 ChIP worked.
13. 3J, line 194-195: heterochromatic genes remaining silent upon macroH2A1 reduction has been observed and the redundant layers of silencing hypothesis has been put forward (this should be cited)
14. It is not clear what the significance is referring to in the figures (B,D) as presented. For example is cpne6.1 really have less mH2A2 in the Lsh KO?
15. 5A: needs a scale bar, not a clear correspondence between ChIP-seq results and chromatin association in 4A, to compare with 5A,E
16. Line 225, fig 5C – peaks not really an appropriate analysis for macroH2A, the need to be using broad domain callers for macroH2A enrichment – like SICER
17. Fig 5GHI are mislabeled in test starting on line 235
18. Fig 5HIG It's not clear what repeats are being analyzed.

Peer review by Matthew Gamble

Reviewer #3:

Remarks to the Author:

In this manuscript, Ni et al thoroughly demonstrate the Lsh is required for appropriate deposition of the histone H2A variant, macroH2A, on chromatin. This is likely the first example of a factor responsible for macroH2A appropriate incorporation of macroH2A. Specifically, the authors identify that repression of GFP-Oct4 expression is coupled with reduced macroH2A occupancy in engineered ES cells and further demonstrate that depletion or deletion of Lsh reduces macroH2A chromatin levels (relative to whole cell levels) and occupancy on chromatin. Importantly, macroH2A depletion leads to similar phenotypes as seen in Lsh KD/KO cells. The authors take their study even further to examine how mutations in Lsh associated with IC4 patients have reduced macroH2A localization in modeled ES cells or in cells derived from a patient. While I have the below suggestions to perhaps strengthen the manuscript, I think overall these are exciting findings.

- 1) After demonstrating reduced expression of GFP-Oct4, the authors go on to investigate what chromatin alterations are accompanying this decrease in expression. Here the authors examine a huge number of histone modifications and some histone variants (which is appreciated) and this is where authors identify a reduction in macroH2A occupancy in an Lsh-ATP dependent manner. I find it very surprising that the authors examined so many modifications, but did not examine other major H2A variants, and specifically H2A.Z. Importantly, H2A.Z has been associated with transcriptional changes, so this seems an obvious choice to examine, especially given the change in canonical H2A levels and macroH2A levels.
- 2) In Figure 3, the authors see increase in expression of a number of loci upon both LSH KO and macroH2A KD. However, there appears to be a greater increase at these locations in the macroH2A KD. Does this suggest an additional remodelers activity for macroH2A incorporation? Or an additional function of macroH2A? It would be nice to discuss these differences.
- 3) In discussion of data presented in Figure 4 (line 216), the authors state that changes are mediated through macroH2A. However, there is no epistasis examined. An informative genetic approach would be to take the Lsh KO and depleted macroH2A in these cell lines to determine if there is any additive effect or if it is truly mediated by macroH2A.
- 4) The inclusion of genomic ChIP-seq data to examine macroH2A localization is great to include.

However, I have some inquiries about these data and analyses:

- a. Are these spike in normalized?
- b. Data in panel B show 80% of genome is unchanged in macroH2A occupancy, is this because at those locations macroH2A is just not normally there, or at low levels? Or are these locations that Lsh does not localize?
- c. In panel F: all genes is nice to include, but data might be more informative to include also the unchanged genes also providing number of genes would be helpful.
- d. Panel. 5H is misleading: total number of peaks in WT vs KO is different, so the numbers are not comparable. I would recommend doing it as number of peaks and then that number is 100% for each and fill it out with alternative filling in the bars.
- e. In the analysis description, the aligning algorithm/tool used is not mentioned
- f. In the analysis description the authors state "Resulting bins were the sum all the single base pair coverage in the 5Kb regions" – these data do not provide single base pair resolution (as something like ChIP-exo or ChIP-nexus would).
- g. All analysis is done in mm9 which is outdated. Using mm10 (which has now been available for ~9 years) would be more helpful to the community.
- 5) The authors move on to model IC4 mutations in mouse ES cells. Interestingly while 2 of the mutants have some effect on GFP-Oct4 expression, 1 has an effect similar to the Lsh ATPase mutation. It is surprising then that the reduction in macroH2A occupancy is similar in these 3 mutants, and especially for 784). Do the authors have an explanation or speculation on these, perhaps complicated, differences?

Minor points:

- 1) In Figure 1, the ChIP experiments only show input. As a control, were IgG control experiments performed to demonstrate background enrichment?
- 2) The authors use FACs to show decreased GFP levels upon Lsh recruitment to their engineered Oct4 allele, however the RTqPCR shows only a 2-fold decrease in GFP levels. Is this due to GFP stability and therefore examining nascent levels over a timecourse might better depict the extent of reduction?
- 3) In figure 1C, there is something labeled "control" but it is not clear what this control is.
- 4) I am surprised to see that in Figure 1, the authors see no change to total Oct4 levels, even when the engineered allele is repressed. Is there no indication of differentiation or morphological changes to the cells?
- 5) In Figure 2A, the "-3000" is out of line with the rest of the numbers
- 6) For experiments in. Figure 3, the authors switch to MEFs and U2OS, Presumably the switch to MEFs is due to acquisition of a Lsh KO, however in data from Figure 4 and beyond it appears the authors have now Lsh KO ES cells. So while I appreciate that this is not a unique event to ES cells, why not include the KO ES cells in these experiments to add continuity to the manuscript?
- 7) For Figure 3J, authors discuss that expression is unchanged from data in reference 28. It would be nice to include these data if available?
- 8) In line 211, the authors state ES, when it should be ES cells (or ESCs)
- 9) For the first half of the manuscript, authors always refer to macroH2A as such, and then around line 215, start modifying to mH2A. This adds unnecessary confusion for readers.
- 10) It is not obvious whether replicates in Figure 5 are independent KO cell lines (independently derived clones) or if they are technical replicates from the same cell line.
- 11) Lines 235-238 are referring to the wrong figure panel
- 12) In Figure 6B it is not clear what V5 and HA are tagging -- include in the panel or figure legend.

Reviewers' comments:

Reviewer #1 (Remarks to the Author):

In this manuscript Ni et al. establish that the LSH chromatin remodeler mediates gene repression, in part, through inducing the deposition of the histone variant macroH2A. In Figure 1, the authors use a chemical genetic approach to induce the recruitment of LSH to a GFP:OCT4 gene report by rapamycin treatment in ES cells and show that LSH recruitment represses the gene about 2 fold over a 5 day period in a manner that requires LSH's ATPase activity. In figure 2 the authors show that macroH2A1 and macroH2A2 recruitment occupancy to the Oct4 reporter increases in a LSH ATPase dependent manner over a 3 day rapamycin treatment. Next, the authors show that Lsh promotes macroH2A1 recruitment to rRNA genes and a variety of other repeat sequences (Fig 3). In figure 4, the authors argue that Lsh loss leads to a global reduction in macroH2A1 deposition using chromatin fractionation and nuclear size as a surrogate. In figure 5, the authors use ChIP-seq to demonstrate loss of macroH2A deposition genome wide. In figure 6, the authors examine the role of an LSH mutation found in a genetic disease, immunodeficiency centromeric instability facial anomalies (ICF), and provide evidence that this mutation affects macroH2A deposition. Finally, in figure 7 demonstrate that human cells from a patient with ICF4 have a reduction in macroH2A1 deposition at repeats.

Major comments:

1. This is an interesting manuscript that makes an important discovery regarding the role of Lsh in mediating macroH2A deposition for the purposes of silencing. Identifying the machinery responsible for macroH2A deposition has been a roadblock for the field and so this paper represents an advance that will be of great interest. However, there are a variety of issues that the authors need to address before the manuscript is suitable for publication

We like to thank the reviewer for the helpful suggestions to improve the manuscript.

2. The manuscript implies that Lsh is a chromatin remodeler capable of depositing macroH2A1 into chromatin, in a manner analogous to how the SWR1 complex mediates the deposition of H2AZ. Another possibility is that Lsh acts upstream of macroH2A deposition through its chromatin remodeling function but that some other factor mediates macroH2A deposition into chromatin. If Lsh is directly responsible for macroH2A deposition, the author should show that Lsh interacts with macroH2A and is selective for macroH2A over other H2A-type histones.

We have now performed CO-IPs as suggested by the reviewer and found that LSH indeed interacts with macroH2A. We show that LSH pulls down macroH2A and vice versa macroH2A pulls down LSH. Furthermore, the reaction is selective, since two LSH mutations found in ICF4 patients (and incapable to deposit macroH2A in the CIP assay) show reduced interaction. In addition, LSH does

not interact with H2A.Z, and H2A.Z occupancy remains unchanged in the CIP assay. These new findings support the notion that LSH is directly responsible for the deposition of macroH2A into chromatin.

These new data is presented in supplementary Fig. 4e, Fig. 4g, h and Fig. 6f, and discussed on pages 8, 12 and 16.

3. Again if Lsh is directly mediating macroH2A deposition, ChIP-seq should be able to show that Lsh and macroH2A localize to the same genomic regions. No Lsh ChIP is presented.

We now examine LSH occupancy by qPCR analysis at sites of high macroH2A occupancy such as repeat sequences and ribosomal genes, and find good concordance of macroH2A and LSH enrichment.

Furthermore, we are now including a genome wide analysis of previously published LSH ChIP-seq data retrieved from the GEO database. We first validated the high throughput seq data using ChIP followed by qPCR analysis, and found good accordance of LSH binding at all examined sites (n=10) comparing sites of strong occupancy with low occupancy. We also report that the part of the genomic compartment showing changes upon LSH depletion exhibits significant higher LSH occupancy. Moreover, the genome wide Pearson correlation of macroH2A enrichment and LSH occupancy (5kb bins) is $R = 0.435$ (untransformed data) and $R = 0.768$ (\log_2 transformed data) indicating medium to strong association between LSH and macroH2A deposition in the genome. These findings are consistent with a direct role for LSH in macroH2A deposition, but it does not exclude that other factors contribute as well to the distribution pattern of macroH2A (as discussed below)

These new data is presented in Fig. 3b, d, supplementary Fig. 5b, d, Fig. 5c-e and supplementary Fig. 8b, c, and discussed on pages 9, 13 and 14.

4. The model presented by the authors is in stark contrast to the kinetic model presented by the Bernstein lab where macroH2A is shown to be deposited everywhere and is specifically removed from locations. Discussion on how/if these two models can coexist needs to be presented.

We assume that multiple pathways, including a LSH dependent pathway and a mechanism based on transcription pruning, act together to shape genome wide macroH2A deposition.

It was previously noted that lower levels of macroH2A are associated with gene-poor sites, suggesting that factors distinct from gene transcription are involved to explain the distribution of macroH2A (61). Please also note point 14, that LSH can change macroH2A deposition at silenced genes independent of transcriptional changes (Supplementary Fig. 6).

The UCSC genome browser snapshot in Supplementary. Fig 8e illustrates how the pathways may co-exist: In the gene-free regions with low LSH occupancy associated with low macroH2A deposition and high LSH binding associated with high macroH2A occupancy, the absence of LSH

maybe a determining factor since the region does not contain genes which could be transcribed. On the other hand, the broad domain with high LSH occupancy includes small regions with low macroH2A enrichment which correspond with the localization of genes. Here, transcriptional pruning may remove macroH2A.

Our model proposes that LSH deposits macroH2A at specific regions of the genome, and transcription prunes and fine tunes the amount of macroH2A at any given genomic site. This is discussed on pages 10, 11, 14 and 18.

5. The authors make the point that the Lsh KO in mice recapitulates many features of ICF, but to my knowledge macroH2A KO mice do not recapitulate features of ICF. This needs to be discussed.

There is some degree of similarity comparing the macroH2A KO phenotype with the *Lsh* KO phenotype. For example, macroH2A or LSH depletion reduces polycomb repression at *Hox* cluster target genes (65, 66); removal of macroH2A increases the efficiency of induced pluripotency (67) and LSH removal enhances cellular plasticity (68); knockdown of macroH2A or LSH in ES cells results in incomplete silencing of pluripotency genes *Oct4*, *Nanog* and *Sox2* (45, 69, 70); knockout of macroH2A2 or LSH in mice results in smaller body size of day 18.5 fetus and macroH2A and *Lsh* KO mice show increased perinatal lethality (30% and 100% respectively), but neither phenotype shows pathology that would point to the cause of death (53, 71); there are reproductive problems in macroH2A knockout mice in C57BL/6 background and LSH knockout mice show defects in male and female germ cell development (18, 53, 72).

Despite some degree of phenotypic resemblance, LSH may not mediate all its function through macroH2A. As pointed out above, LSH knockout mice exhibit more severe lethality, and LSH deficient cells still have a remainder of macroH2A deposition, suggesting that they do not share a completely congruent pathway.

With respect to the ICF4 syndrome, to our knowledge genomic instability, immunodeficiency and neurologic function, three important features in the ICF syndrome have not yet been assessed in the macroH2A KO mice. The discussion is included on pages 19 and 20.

6. MacroH2A is not only associated with transcriptional silencing and not only associated with constitutive heterochromatin (as expressed on line 62). It is also found in facultative heterochromatin and euchromatin. MacroH2A is a family of variants, macroH2A1.1, macroH2A1.2 and macroH2A2. MacroH2A1.1 is distinct in both function (it plays a positive role in transcription) and localization (It resides in euchromatic regions). Which macroH2A1 variants are expressed in these cells should be quantified (typically ES cells do not express much macroH2A1.1). If so, this should be clarified so as not to confuse the field about the function of these distinct histone variants.

We have now examined the expression of macroH2A1 isoforms in all cell types used in this study. We report that macroH2A1.2 is the prevalent isoform in our study. Our results are consistent with

previous studies (13, 45-47). We have now included more information about the isoforms and their functions (9-14), and added expression information in Supplementary Fig. 4a and discussed on pages 3, 7, 8 and 9.

7. Fig 1: There is a disconnect between where LSH is recruited and where macroH2A1 deposition occurs. I can see how macroH2A might not be able to be recruited directly at the site of LSH recruitment, but why would it be asymmetrical – macroH2A1 is only getting recruited to the gene body, not a similar distance upstream? This needs to be discussed.

An interesting observation, we can only speculate about the cause of asymmetry. Most remodelers anchor at a fixed position on the histone octamer and show unidirectional sliding along DNA (4, 24). The manner of LSH (or a LSH complex) contacting the nucleosome may give LSH a certain orientation. The translocase domain of LSH may impose unidirectional sliding along DNA and directional nucleosome mobilization and histone exchange. This is discussed on page 17.

8. There are no statistics in any of the critical ChIPs demonstrating the significance of change in factor recruitment (Fig 2, Fig 6E).

Fig. 2 and Fig. 6e include now statistical analysis as indicated by asterisks.

9. In Fig 4, the authors suggest that there is less chromatin associated macroH2A while total protein levels remain the same in the Lsh KO. But it's not much reduced in chromatin, so where is all of that macroH2A going and how is that result consistent with the macroH2A ChIP-seq data shown in Fig 5?

The total macroH2A protein amount comparing LSH deficient to wild type cells does not change. However, the amount of macroH2A in the chromatin fraction is reduced in LSH deficient cells by about 40 % compared to wild type cells, which is consistent with the model that less macroH2A is incorporated into chromatin. We have not tracked the incorporated proportion in cytosolic or the nuclear fraction since they are highly diluted, and we do not relate the chromatin fraction to the total protein amount, but focus on the difference between *Lsh* KO and WT. Similar observations have been made for H2A.Z chromatin association in cells deficient of the H2A.Z exchange factor EP400 (54). This is discussed on page 11.

The relation to Fig. 5 is further explained under point 15 below.

Minor

comments:

10. Fig 1C and 6C : ChIP should be expressed as % input rather than fold enrichment (which is ambiguous) as done in 1F

We have changed the figures according to the reviewer's suggestion.

11. no comment was made under number 11

12. Fig 2F: What is the positive control to suggest the H1.2 ChIP worked.

We are now displaying a positive control (major satellite sequences) and negative control (Gapdh) in the inlet of Fig. 2f used for H1.2 ChIPs validation in accordance with previous publications (41, 42) as stated on page 7.

13. 3J, line 194-195: heterochromatic genes remaining silent upon macroH2A1 reduction has been observed and the redundant layers of silencing hypothesis has been put forward (this should be cited)

We cited accordingly previous work (9, 51-53) and the redundant layers of silencing hypothesis on pages 10 and 11.

14. It is not clear what the significance is referring to in the figures (B,D) as presented. For example is cpne6.1 really have less mH2A2 in the Lsh KO?

Repeat regions (Fig. 3b, d) have significant changes in macroH2A occupancy and in gene expression. Since transcriptional pruning is a mechanism that reduces macroH2A at highly transcribed genes, we wanted to examine whether changes of macroH2A in LSH deficient cells depend on transcriptional changes.

The Fig. 3j (now supplementary Fig. 6h) illustrate examples, in which the reduction of macroH2A enrichment in LSH deficient cells does not depend on transcriptional changes, because these genes are silent in wild type and in *Lsh* knockout cells. In other words, the loss of macroH2A at these sites is associated with a lack of LSH, and not with transcriptional changes.

The previous PCR primer location was chosen before our ChIP-seq data was available, and was by chance located at a region within the Cpne6 gene with low macroH2A occupancy in wild type cells. We now designed another primer and show that the decrease in macroH2A1 and macroH2A2 at the gene body of Cpne6 is significant. In addition, we show the UCSC genome browser views of the ChIP-seq data of the examined genes. The snapshots visualize the primer locations and display the reduction of macroH2A over all examined genes. These examples illustrate that a change in macroH2A can occur without transcriptional changes. We present the data in Supplementary Fig. 6b-h and discuss this on pages 10 and 13.

15. 5A: needs a scale bar, not a clear correspondence between ChIP-seq results and chromatin association in 4A, to compare with 5A,E

We added the scale bar in Fig. 5a.

The Chip-seq data suggest that macroH2A deposition is reduced, but not completely absent in LSH deficient cells. Certain regions show a substantial reduction of macroH2A deposition (UCSC genome browser view, Fig. 5a) and analysis of previously published macroH2A peaks show a considerable decrease in macroH2A enrichment (Fig. 5e, now Fig. 5g). The global analysis shows that the number of macroH2A peaks and broad macroH2A domains are reduced by about 50% and

30%, respectively, comparing LSH deficient cells with wild type cells (Fig. 5f, h). Fig. 4a examines the amount of LSH protein associated with chromatin, which is reduced by about 40% comparing the Lsh deficient sample with wild type sample. This estimate corresponds reasonably well with the estimate based on the global ChIP-seq analysis. This is discussed on page 14.

16. Line 225, fig 5C – peaks not really an appropriate analysis for macroH2A, the need to be using broad domain callers for macroH2A enrichment – like SICER

We used MACS software because it had been previously used for macroH2A ChIP-seq analysis (51, 59). We have now also conducted a broad domain analysis using Epic2 software, an update to SICER (62, 63). The results corroborate our conclusions from the peak analysis. We present results of both methods and show that significant decreases of macroH2A enrichment in small regions (peaks, MACS) and broad domains (Epic2) occur in LSH deficient cells.

The new results of Epic2 are presented in Fig. 5h and discussed on page 14.

17. Fig 5GHI are mislabeled in test starting on line 235

We corrected the errors.

18. Fig 5HIG It's not clear what repeats are being analyzed.

We derived the repeat sequences from the RepeatMasker data base (<http://www.repeatmasker.org/>), and provide a list of repeat sequences with expression data in the source data file.

Peer review by Matthew Gamble

Reviewer #2 (Remarks to the Author):

In this manuscript, Ni et al thoroughly demonstrate the Lsh is required for appropriate deposition of the histone H2A variant, macroH2A, on chromatin. This is likely the first example of a factor responsible for macroH2A appropriate incorporation of macroH2A. Specifically, the authors identify that repression of GFP-Oct4 expression is coupled with reduced macroH2A occupancy in engineered ES cells and further demonstrate that depletion or deletion of Lsh reduces macroH2A chromatin levels (relative to whole cell levels) and occupancy on chromatin. Importantly, macroH2A depletion leads to similar phenotypes as seen in Lsh KD/KO cells. The authors take their study even further to examine how mutations in Lsh associated with IC4 patients have reduced macroH2A localization in modeled ES cells or in cells derived from a patient. While I have the below suggestions to perhaps strengthen the manuscript, I think overall these are exciting findings.

We like to thank the reviewer for the helpful suggestions to improve the manuscript.

1) After demonstrating reduced expression of GFP-Oct4, the authors go on to investigate what chromatin alterations are accompanying this decrease in expression. Here the authors examine a huge number of histone modifications and some histone variants (which is appreciated) and this is where authors identify a reduction in macroH2A occupancy in an Lsh-ATP dependent manner. I find it very surprising that the authors examined so many modifications, but did not examine other major H2A variants, and specifically H2A.Z. Importantly, H2A.Z has been associated with transcriptional changes, so this seems an obvious choice to examine, especially given the change in canonical H2A levels and macroH2A levels.

We are now including an analysis of the important histone variant H2A.Z. CIP analysis shows no changes in H2A.Z enrichment upon LSH recruitment. Moreover, our revision also includes the data showing that LSH co-immunoprecipitated with macroH2A1 and macroH2A2, but not with H2A.Z. This suggests that LSH shows some specificity with respect to histone variants and that LSH selectively changes macroH2A, but not H2A.Z occupancy.

The new data is presented in Supplementary Fig. 4e and Fig. 4g and discussed on pages 8 and 12.

2) In Figure 3, the authors see increase in expression of a number of loci upon both LSH KO and macroH2A KD. However, there appears to be a greater increase at these locations in the macroH2A KD. Does this suggest an additional remodelers activity for macroH2A incorporation? Or an additional function of macroH2A? It would be nice to discuss these differences.

We now discuss this observation and the possibility that other chromatin remodelers and factors may contribute to macroH2A deposition on pages 14 and 18.

3) In discussion of data presented in Figure 4 (line 216), the authors state that changes are mediated through macroH2A. However, there is no epistasis examined. An informative genetic approach would. Be

to take the Lsh KO and depleted macroH2A in these cell lines to determine if there is any additive effect or if it is truly mediated by macroH2A.

We have now added the suggested experiment and depleted macroH2A1+2 in *Lsh* KO cells. We observe that *Lsh* KO does not further change the nuclear size once macroH2A1+2 is depleted. In addition, we examined transcriptional re-activation using the same approach and found no further additive effect by LSH depletion once macroH2A1+2 is removed. This data supports the notion that changes in transcription and in nuclear size are likely mediated through macroH2A.

The new data is now presented in Fig. 3g-i, Supplementary Fig. 5e-g, Fig. 4b, c and Supplementary Fig. 7b-f, and discussed on pages 10, 11 and 12.

4) The inclusion of genomic ChIP-seq data to examine macroH2A localization is great to include. However, I have some inquiries about these data and analyses:

a. Are these spike in normalized?

The peak data has been normalized for input control with each individual ChIP sample normalized to its individual input (supplementary Table 2 lists all samples). The normalization is described on page 28.

b. Data in panel B show 80% of genome is unchanged in macroH2A occupancy, is this because at those locations macroH2A is just not normally there, or at low levels? Or are these locations that Lsh does not localize?

We discuss now that both assumptions are probably correct: 80% of the genome which is unchanged in macroH2A occupancy shows also significantly lower macroH2A occupancy in wild type samples. In addition, we have analyzed previously published LSH ChIP-seq data and find that the 80% of the genome also show significantly lower LSH occupancy further corroborating our findings, which suggest that LSH is critical for macroH2A deposition.

This information is now presented in Supplementary Fig. 8b, c and discussed on page 13.

c. In panel F: all genes is nice to include, but data might be more informative to include also the unchanged genes also providing number of genes would be helpful.

We have now changed the figure accordingly (now Fig. 5i).

d. Panel. 5H is misleading: total number of peaks in WT vs KO is different, so the numbers are not comparable. I would recommend doing it as number of peaks and then that number is 100% for each and fill it out with alternative filling in the bars.

We modified Fig. 5h according to the reviewer's suggestion, but moved it to the supplement because of space restriction. The figure is now shown in Supplementary Fig. 8d.

e. In the analysis description, the aligning algorithm/tool used is not mentioned

The reads were trimmed using Cutadapt 1.18 to remove adapters and low-quality flanking regions. These reads were aligned to the mouse genome (mm9) using Bowtie2 2.2.6. The information has been added on page 28.

f. In the analysis description the authors state “Resulting bins were the sum all the single base pair coverage in the 5Kb regions” – these data do not provide single base pair resolution (as something like ChIP-exo or ChIP-nexus would).

We corrected the error in the wording.

g. All analysis is done in mm9 which is outdated. Using mm10 (which has now been available for ~9 years) would be more helpful to the community.

We agree that MM9 is in many ways outdated. However, regarding "Expression and Regulation" of genes MM9 has more features (34 features including histones, histone modifications, and chromatin accessibility) as opposed to MM10 (only 9 features) that a reader may find useful for comparison.

5) The authors move on to model ICF4 mutations in mouse ES cells. Interestingly while 2 of the mutants have some effect on GFP-Oct4 expression, 1 has an effect similar to the Lsh ATPase mutation. It is surprising then that the reduction in macroH2A occupancy is similar in these 3 mutants, and especially for 784). Do the authors have an explanation or speculation on these, perhaps complicated, differences?

All three ICF4 mutants show an impairment in GFP repression compared to wild type LSH, albeit to a different degree. About 87% of wild type cells show significant repression upon rapamycin treatment. In contrast, only 1% of cells expressing mutant S 745 Rfs*4 show GFP repression, and only 12% of mutant Q 682 R and 9% of mutant L 784 del cells show GFP repression. We hypothesize, that each mutation may have a slightly different effect on LSH function. For example, mouse mutants Q 682 R (Q 699 R in human) and L 784 del (L 801 del in human), but not mutant S 745 Rfs*4 (S 762 Rfs*4 in human), show impaired interaction with macroH2A (Fig. 6f). The latter mutation may interfere with ATPase function or chromatin remodeling. This is now clarified and discussed on pages 15, 16, 19 and 20.

Minor

points:

1) In Figure 1, the ChIP experiments only show input. As a control, were IgG control experiments performed to demonstrate background enrichment?

IgG controls had been performed and are now displayed in the figure 1.

2) The authors use FACs to show decreased GFP levels upon Lsh recruitment to their engineered Oct4 allele, however the RTqPCR shows only a 2-fold decrease in GFP levels. Is this due to GFP stability and therefore examining nascent levels over a timecourse might better depict the extent of reduction?

We have now improved our experimental conditions for RT-PCR analysis, which may have previously picked up genomic DNA (since the GFP vector does not contain any intron suitable for primer design). We designed new primers and treated the RNA samples with RNase-free DNase to reduce the chance of DNA contamination. We observed now a reduction of GFP mRNA to about 20% compared to the starting GFP expression (100%). This corroborates the reduction of RNA Poly II binding over the gene body which is reduced to about 17% (Fig. 1f). The improved time kinetic of GFP mRNA analysis is depicted in Supplementary Fig. 3b.

3) In figure 1C, there is something labeled “control” but it is not clear what this control is.

The control means engineered ES cells treated with rapamycin which only expressed DNA binding fusion protein ZnF-FKBP-HA, but not the LSH expression vector (LSH-FRB-V5). This information is added to the Figure legend. We also display this information in the Western blot analysis (Fig. 1b).

4) I am surprised to see that in Figure 1, the authors see no change to total Oct4 levels, even when the engineered allele is repressed. Is there no indication of differentiation or morphological changes to the cells?

Only one *Oct4* allele is genetically modified and cannot express OCT4, whereas the other allele still expresses wild type OCT4 and maintains the undifferentiated ES cell state (34, 36). It was previously reported that treatment with rapamycin does not induce differentiation of engineered ES cells (34, 36). We now show that there is no evidence of morphologic change and report that mRNA for the stem cell genes *Oct4*, *Nanog* and *Sox2* remain unchanged, and OCT4 protein levels remain the same upon rapamycin treatment.

The new data is shown in Fig. 1e, Supplementary Fig. 3a, c-f and discussed on pages 5 and 6.

5) In Figure 2A, the “-3000” is out of line with the rest of the numbers

We changed the Figure accordingly.

6) For experiments in Figure 3, the authors switch to MEFs and U2OS, Presumably the switch to MEFs is due to acquisition of a Lsh KO, however in data from Figure 4 and beyond it appears the authors have now Lsh KO ES cells. So while I appreciate that this is not a unique event to ES cells, why not include the KO ES cells in these experiments to add continuity to the manuscript?

We have tried to include several different cell types in our study including ES cells, MEFs, U2OS cells and human patient LCL cells. A main reason for using MEFs and U2OS cells in Fig. 3 and Supplementary Fig. 5 is the higher amount of macroH2A in differentiated cells compared to ES cells,

as previously reported (13, 45-47). We have examined the macroH2A1 isoforms for all cells used in this study, which illustrate the low macroH2A1 content in ES cells.

These new data is shown in Supplementary Fig. 4a, and discuss the results on pages 7, 8 and 9.

7) For Figure 3J, authors discuss that expression is unchanged from data in reference 28. It would be nice to include these data if available?

The data is now displayed in Supplementary Fig. 6a.

8) In line 211, the authors state ES, when it should be ES cells (or ESCs)

We corrected the sentence accordingly.

9) For the first half of the manuscript, authors always refer to macroH2A as such, and then around line 215, start modifying to mH2A. This adds unnecessary confusion for readers.

We uniformly use now macroH2A throughout the manuscript.

10) It is not obvious whether replicates in Figure 5 are independent KO cell lines (independently derived clones) or if they are technical replicates from the same cell line.

The replicates are from different cell lines derived from different individual embryos (three for each genotype). We added this information in figure legend, and on pages 12 and 31.

11) Lines 235-238 are referring to the wrong figure panel

We corrected this error.

12) In Figure 6B it is not clear what V5 and HA are tagging -- include in the panel or figure legend.

We added this information in the figure legend and change the format of Fig. 6b for clarification.

Reviewers' Comments:

Reviewer #2:

Remarks to the Author:

Ni and colleagues did a commendable job in responding to my prior critique (and that of the 2nd reviewer). I believe the data is exciting and of interest to a broad readership. I have one remaining comment. I would ask that the authors be specific throughout the text (especially in the discussion) and refer to either macroH2A1.1, macroH2A1.2 or macroH2A2 where appropriate. Given that the cell line used in the study primarily expresses macroH2A1.2, the authors should avoid overly generalizing their findings to the other macroH2A isoforms, as these forms (and especially macroH2A1.1) have distinct biological functions.

Matthew Gamble

Reviewer #3:

Remarks to the Author:

Ni et al present a revised manuscript that is strengthened in its demonstration of Lsh regulating deposition of the histone H2A variant macroH2A. In my opinion, the additional experiments, including the co-IP experiments and macroH2A depletion in Lsh KO cells have strengthened the initial manuscript.

However, I have one remaining concern which was originally stated in the first review, which is that analysis was performed using the 10-year replaced mm9 genome build. While the authors agree with this, they still have not analyzed their data in the most up to date genome annotation, mm10.

The authors argue that there are more features in mm9, however, these can be easily converted to mm10 and therefore could be applied.

We like to thank the reviewers for the helpful suggestions to improve the manuscript. Please see below for our point-by-point response to the reviewers' comments and concerns.

Reviewer #2 (Remarks to the Author):

Ni and colleagues did a commendable job in responding to my prior critique (and that of the 2nd reviewer). I believe the data is exciting and of interest to a broad readership. I have one remaining comment. I would ask that the authors be specific throughout the text (especially in the discussion) and refer to either macroH2A1.1, macroH2A1.2 or macroH2A2 where appropriate. Given that the cell line used in the study primarily expresses macroH2A1.2, the authors should avoid overly generalizing their findings to the other macroH2A isoforms, as these forms (and especially macroH2A1.1) have distinct biological functions.

R: We have now revised the manuscript and refer to either macroH2A1.1, macroH2A1.2 or macroH2A2 where appropriate.

Reviewer #3 (Remarks to the Author):

Ni et al present a revised manuscript that is strengthened in its demonstration of Lsh regulating deposition of the histone H2A variant macroH2A. In my opinion, the additional experiments, including the co-IP experiments and macroH2A depletion in Lsh KO cells have strengthened the initial manuscript.

However, I have one remaining concern which was originally stated in the first review, which is that analysis was performed using the 10-year replaced mm9 genome build. While the authors agree with this, they still have not analyzed their data in the most up to date genome annotation, mm10. The authors argue that there are more features in mm9, however, these can be easily converted to mm10 and therefore could be applied.

R: We have now analyzed our data in mm10. We revised Fig. 5, Supplementary Fig. 6 and Supplementary Fig. 8. We also changed the manuscript and the Geo submission accordingly.